# Apparent nonlinear damping triggered by quantum fluctuations

**Mario F. Gely** [1,3] ✉, **Adrián Sanz Mora**[1], **Shun Yanai**[1,4,5], **Rik van der Spek**[1], **Daniel Bothner** [1,2] & **Gary A. Steele** [1]

Nonlinear damping, the change in damping rate with the amplitude of oscillations plays an important role in many electrical, mechanical and even biological oscillators. In novel technologies such as carbon nanotubes, graphene membranes or superconducting resonators, the origin of nonlinear damping is sometimes unclear. This presents a problem, as the damping rate is a key figure of merit in the application of these systems to extremely precise sensors or quantum computers. Through measurements of a superconducting resonator, we show that from the interplay of quantum fluctuations and the nonlinearity of a Josephson junction emerges a power-dependence in the resonator response which closely resembles nonlinear damping. The phenomenon can be understood and visualized through the flow of quasi-probability in phase space where it reveals itself as dephasing. Crucially, the effect is not restricted to superconducting circuits: we expect that quantum fluctuations or other sources of noise give rise to apparent nonlinear damping in systems with a similar conservative nonlinearity, such as nano-mechanical oscillators or even macroscopic systems.

Amplitude-dependent (nonlinear) damping is ubiquitous in nature. It was famously described mathematically by van der Pol[1] in the context of his work on vacuum tube circuits[2]. Now, it is used to describe the physics of a diverse set of systems, such as the rolling of ships in waves[3] or the nervous system[4]. It has attracted recent interest due to its appearance in novel experimental platforms such as nanoscale ferromagnets[5], superconducting circuits[6–9] and nanoelectromechanical systems (NEMS)[10–13] made for example from carbon nanotubes, graphene[14,15] or superconducting metal[16]. In some of these systems the nonlinearity is well explained[17–20]. Most notably the saturation of two-level systems in the environment can cause negative nonlinear damping: the damping rate decreases as the power injected into the system increases[6,8,9,16]. But the origin of an increase in damping with power in certain NEMS[11–14] or superconducting resonators[7] remains speculative. Understanding the origin of nonlinear damping in

some of these systems is critical due to the importance of their energy damping rates in applications such as NEMS based mass sensing[21] or spectrometry[22], as well as quantum-limited amplification[23,24] in superconducting quantum computers[25].

We study a dephasing effect in superconducting circuits[26], which phenomenologically appears as nonlinear damping when measuring the resonant response of a resonator. Central to the observed physics is the nonlinearity induced by a Josephson junction: that the resonance frequency varies with the oscillation amplitude, which can be further approximated as a Duffing or Kerr nonlinearity[27]. For this reason, the phenomena discussed here are applicable to all systems featuring a similar nonlinearity in their resonance frequency, for example the carbon-nanotubes mentioned above, or even a macroscopic mechanical pendulum. We focus on the regime where this nonlinearity is small, as in Josephson parametric amplifiers[24], rather than the single-photon

[1]Kavli Institute of NanoScience, Delft University of Technology, PO Box 5046, 2600 GA Delft, The Netherlands. [2]Physikalisches Institut, Center for Quantum Science (CQ) and LISA+, University of Tübingen, Auf der Morgenstelle 14, 72076 Tübingen, Germany. [3]Present address: Clarendon Laboratory, Department of Physics, University of Oxford, Parks Road, Oxford OX1 3PU, UK. [4]Present address: Institute for Quantum Computing, University of Waterloo, 200 University Avenue West, Waterloo, ON N2L 3G1, Canada. [5]Present address: Department of Physics and Astronomy, University of Waterloo, 200 University Avenue West, Waterloo, ON N2L 3G1, Canada. ✉e-mail: mario.gely@physics.ox.ac.uk

nonlinear regime used to construct artificial atoms in circuit quantum electrodynamics[27]. Because of their small nonlinearity, such systems are often thought to be completely described by the classical Kerr oscillator[7,28].

Here however, we report on an effect that is not expected from the classical Kerr oscillator. More specifically, we present a phenomenon triggered by the interplay between the quantum noise and Kerr anharmonicity of the oscillator, which closely resembles nonlinear damping in the steady-state response of the oscillator. The apparent nonlinear damping is first experimentally characterized by probing the frequency response of the resonant circuit. Our observations are then accurately described by a quantum theory of a damped driven Kerr oscillator devoid of ad hoc nonlinear damping, but which takes into account the effect of quantum noise. Moreover, focusing on an oscillator steady-state below its bistability threshold, a Gaussian state approximation[29] allows us to demonstrate that, in a close vicinity of the resonance, the expected amplitude of oscillations is akin to that of a driven classical Kerr oscillator with nonlinear damping. Finally, we provide an intuitive picture in which the phenomenon can be understood as the oscillator experiencing dephasing induced by its own photon shot noise.

## Results

### Experimental setup

The circuit used in this experiment (Fig. 1) is constructed from an inductor, capacitor and superconducting quantum interference device, or SQUID. The SQUID is flux-biased to its sweet spot (integer flux quantum), and behaves as a single Josephson junction[30]. The junction induces an anharmonicity of strength $K = 2\pi \times 80$ kHz five orders of magnitude smaller than the resonance frequency $\omega_r = 2\pi \times 5.17$ GHz. The cosine potential of the junction is accurately described in this limit $K \ll \omega_r$ by the Kerr effect in the Hamiltonian[31]

$$\hat{H} = \hbar \left( \omega_r - \underbrace{\frac{K}{2}\hat{a}^\dagger \hat{a}}_{Kerr} - \frac{K}{2} \right) \hat{a}^\dagger \hat{a} ,  \qquad (1)$$

where $\hat{a}$ is the annihilation operator for photons in the circuit. Intuitively, the junction is acting as an inductor, with an inductance which increases with the number of photons $\hat{a}^\dagger \hat{a}$ in the circuit. As a consequence, the resonance frequency of the circuit is lowered with each added photon, labeled as the Kerr term in Eq. (1).

The circuit undergoes internal damping, losing energy at a rate $\kappa_{int} = 2\pi \times 186$ kHz. This is typically due to losses in the different dielectric materials traversed by the electric fields[32]. Additionally, the circuit is coupled to a transmission line, through which we drive the circuit with a microwave signal. Conversely, the transmission line leads to energy leaking out of the circuit, which is characterized by an external damping rate $\kappa_{ext} = 2\pi \times 2.1$ MHz. As a consequence, the total damping rate and spectral linewidth $\kappa = \kappa_{int} + \kappa_{ext}$ is much larger than the shift in resonance frequency $K$ due to an added photon: $\kappa \gg K$. The circuit is thus far from the regime of superconducting qubits[27]. We will call it a Kerr oscillator and first attempt to describe its behavior following the classical equation for the steady-state amplitude of its oscillations $a$

$$\left( i\Delta - iK|a|^2 + \frac{\kappa}{2} \right) a = \epsilon . \qquad (2)$$

Here $\Delta = \omega_r - \omega_d$ is the detuning of the driving frequency $\omega_d$ to the resonance frequency $\omega_r$, and the strength of the drive $\epsilon = \sqrt{\kappa_{ext}P_{in}/(2\hbar\omega_r)}$ is given by $P_{in}$ the power of the drive impinging on the device.

The circuit is made by patterning a thin film of sputtered molybdenum–rhenium alloy on silicon, and subsequently fabricating the aluminum/aluminum-oxide tunnel junctions (see "Methods"). The device is thermally anchored to the ~20 milliKelvin stage of a dilution refrigerator, and the input (output) microwave wiring is attenuated (isolated) to lower its microwave mode temperature, such that the average number of photons $n_{th}$ excited by thermal energy is negligible. The transmission coefficient $S_{21} = 1 - \kappa_{ext}a/(2\epsilon)$ is then measured using a vector network analyzer (VNA) for varying microwave power (see Fig. 2a).

### Experimental data

We note an increase in both the detuning $\Delta_{min}$ which minimizes transmission, and the value of the minimum $\text{Min}|S_{21}|$. The classical prediction $\Delta_{min} = K\alpha^2$ resulting from Eq. (2)—where $\alpha = 2\epsilon/\kappa$ is the expected maximum amplitude—accurately matches the shift of the resonance. However, by plugging the maximum amplitude $\alpha$ into the expression for $S_{21}$, we obtain a constant value for $\text{Min}|S_{21}| = |1 - \kappa_{ext}/\kappa|$ (dashed line in Fig. 2a), which disagrees with the measurement.

In a classical approach to the problem, a power-dependence of the internal damping rate therefore has to create this change. Since $\kappa_{ext}$ is determined by the geometry of the circuit, it should remain unchanged by the power of the drive. For $\kappa_{ext}/\kappa$ to vary and produce the observed change in $\text{Min}|S_{21}| = |1 - \kappa_{ext}/(\kappa_{ext} + \kappa_{int})|$, the internal damping should increase as the drive power increases. At the highest drive power for which data is displayed ($P_{in} = -124$ dBm), the internal damping rises to $2\pi \times 255$ kHz. We note that this power lies below the bistability threshold (see Supplementary Notes 3 and 4F). Such nonlinear damping can be included in the model of Eq. (2) through $\kappa_{int} \rightarrow \kappa_{int}^{nl} = \kappa_{int} + \gamma|a|^2$. We fit a solution of the resulting equation to the data (see Methods), observing good agreement (Fig. 2) for $\gamma = 2\pi \times 5.02$ kHz.

While providing an accurate model for our observations, adding ad-hoc nonlinear damping offers no explanation as to the physical mechanism underlying the effect. Usually, the most prominent source

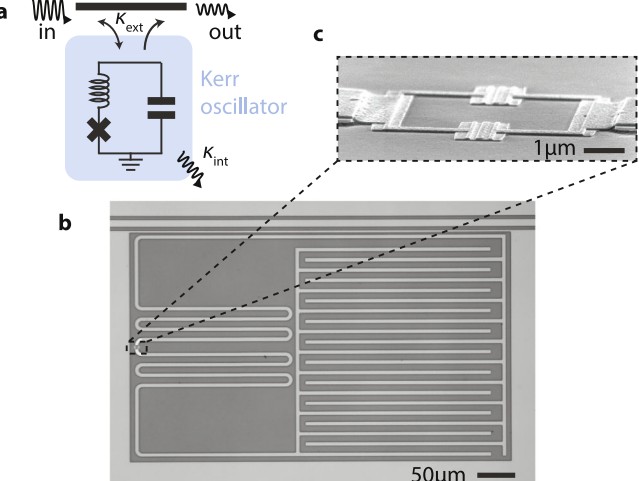

**Fig. 1 | Superconducting Kerr oscillator circuit. a** The Kerr oscillator is constructed from an inductor, a capacitor and a SQUID (which behaves as and is depicted by a single Josephson junction), and is side-coupled to a transmission line with a coupling rate $\kappa_{ext}$. The circuit undergoes internal damping at a rate $\kappa_{int}$. **b** Optical micrograph of the device, where light gray corresponds to superconducting molybdenum-rhenium, and dark gray to the insulating silicon substrate. An interdigitated capacitor on the right is connected to a meandering inductor on the left. The circuit couples to a transmission line (coplanar waveguide) at the top. **c** Scanning electron micrograph of the SQUID: two aluminum/aluminum-oxide Josephson junctions connected in parallel. As the flux threading the SQUID is fixed, it effectively behaves in this context as a single junction.

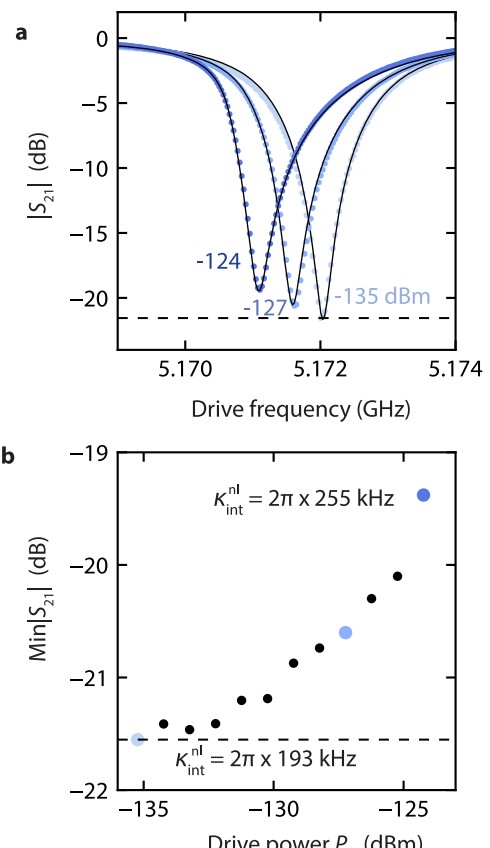

**Fig. 2 | Observation of a resonator steady-state response suggesting nonlinear damping. a** Measured transmission magnitude $|S_{21}|$ (dots) for different drive powers. While the shift in resonance frequency is expected from a classical analysis of the damped driven Kerr oscillator using Eq. (2), $\text{Min}|S_{21}|$ is expected to remain constant (dashed line). **b** Measured $\text{Min}|S_{21}|$ (dots) as a function of drive power. Eq. (2) yields $\text{Min}|S_{21}| = |1 - \kappa_{\text{ext}}/(\kappa_{\text{int}} + \kappa_{\text{ext}})|$, suggesting a damping rate which increases with power $\kappa_{\text{int}} \to \kappa_{\text{int}}^{\text{nl}}(|a|)$ from $\kappa_{\text{int}}^{\text{nl}} = 2\pi \times 193$ kHz to $\kappa_{\text{int}}^{\text{nl}} = 2\pi \times 255$ kHz. Indeed, adding nonlinear damping $\kappa_{\text{int}}^{\text{nl}}(|a|)$ to Eq. (2) leads to theoretical predictions (solid lines in **a**) in good agreement with the data. At the three highlighted points, the expectation values of photon number $|a|^2$ (where the minimum of $|S_{21}|$ is achieved) are 1.1, 6.8 and 13.2.

of nonlinear damping in superconducting circuits is the saturation of two-level systems (TLSs) in the environment[6,8,9]. However, with increasing driving power, the saturation of TLSs will result in a decrease of the internal damping rate, while we observe the opposite. Here, we show that in our system this nonlinear damping behavior can be explained purely by dephasing triggered by the joint action of the intrinsic quantum noise and the Kerr anharmonicity of the oscillator.

## Quantum mechanical simulation

We first show that approaching the problem quantum mechanically, without adding nonlinear damping, perfectly describes our measurements. The effect of quantum noise is included in the model through the steady-state Lindblad equation

$$i\left[\hat{H}/\hbar - \omega_{\text{d}}\hat{a}^\dagger\hat{a} + i\epsilon(\hat{a}^\dagger - \hat{a}), \hat{\rho}\right] = \kappa\left(2\hat{a}\hat{\rho}\hat{a}^\dagger - \hat{\rho}\hat{a}^\dagger\hat{a} - \hat{a}^\dagger\hat{a}\hat{\rho}\right)/2, \quad (3)$$

where $\hat{\rho}$ is the density matrix describing the steady-state of the oscillator. By numerically solving this equation for varying drive strengths and frequencies, the resulting amplitude $\langle\hat{a}\rangle = \text{Tr}(\hat{a}\hat{\rho})$ is used to obtain $S_{21}$. With only the circuit parameters as free variables, and notably a constant value for the internal damping, this model is fitted to all $S_{21}$ traces (see "Methods" and Supplementary Note 2), revealing excellent agreement to the data (Fig. 3). Note that we recently became

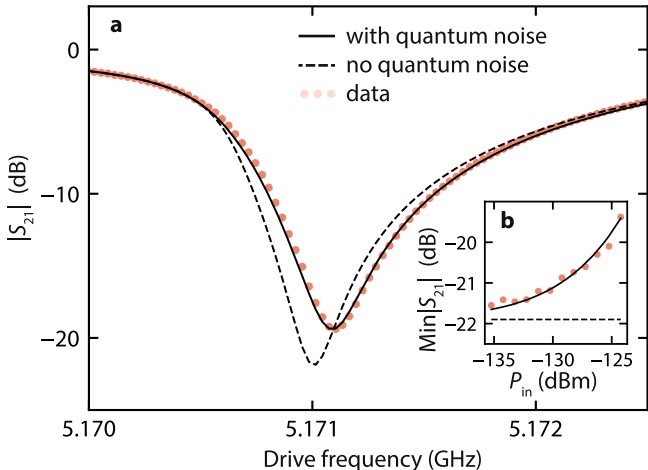

**Fig. 3 | Apparent nonlinear damping triggered by quantum noise.** The experimental results (dots) are compared here to a model with and without quantum noise (full and dashed line respectively). **a** Example of experimental and theoretical $|S_{21}|$ at the input power $P_{\text{in}} = -124$ dBm. **b** As the power varies the model without quantum noise fails to capture the power-dependent depth of the response, which is accurately reproduced when quantum noise is introduced.

aware of an analytical solution to this Lindblad equation[33,34], which may have simplified our approach.

In Fig. 3, we compare this quantum model to the classical model: the solution to Eq. (2), which features neither nonlinear damping nor quantum noise. The only difference between the quantum model—which predicts the increase in $\text{Min}|S_{21}|$—and that of Eq. (2)—which predicts a constant $\text{Min}|S_{21}|$—lies in the value of the commutator $[\hat{a}, \hat{a}^\dagger]$. In fact, by taking the trace $\text{Tr}(\hat{a} \cdot)$ of Eq. (3), and assuming the amplitude to be a complex number $\hat{a} \to a$ such that $[a, a^*] = 0$, we arrive at Eq. (2). Quantum noise can therefore lead to the entirety of the change in $\text{Min}|S_{21}|$. Thermal noise could lead to a similar effect, but is expected to be negligible in our experiment (see Supplementary Note 7).

Beyond describing the data, this model can lead to a nonlinear damping equation for the expectation value $\langle\hat{a}\rangle$. In the Supplementary Note 4, we derive an analytical formula that captures the behavior of the steady-state response that is in good agreement with the numerical simulations. We find that in a resonance scenario, whenever the Kerr effect and thermal noise have only a perturbative effect on the system, the corresponding steady-state expectation value for the amplitude $\langle\hat{a}\rangle$ matches that of a classical non-linearly damped driven classical Kerr oscillator. That is, the steady-state amplitude is ruled by an equation resembling Eq. (2), but where the quantum and thermal noise lead to a nonlinear damping coefficient $\kappa + \gamma|\langle\hat{a}\rangle|^2$ with

$$\gamma = \frac{4K^2}{\kappa}\left(n_{\text{th}} + \frac{1}{2}\right). \quad (4)$$

Here the familiar $+\frac{1}{2}$ stems from quantum noise, which has the same effect as half a quantum of thermal noise. The fact that the nonlinear damping model and the quantum model are both able to describe our measurements is therefore not coincidental: while there is no microscopic process leading to nonlinear damping (i.e., loss of energy), there is apparent nonlinear damping in the equation for $\langle\hat{a}\rangle$ when accounting for the presence of quantum noise. Similar results were derived for the classical[35] and quantum[36] spectrum of undriven oscillators, and also in work studying the spectrum of a probe field in the presence of a strong pump field[37]. The difference here is that we are instead interested in the power dependence of the scattering parameter $S_{21}$.

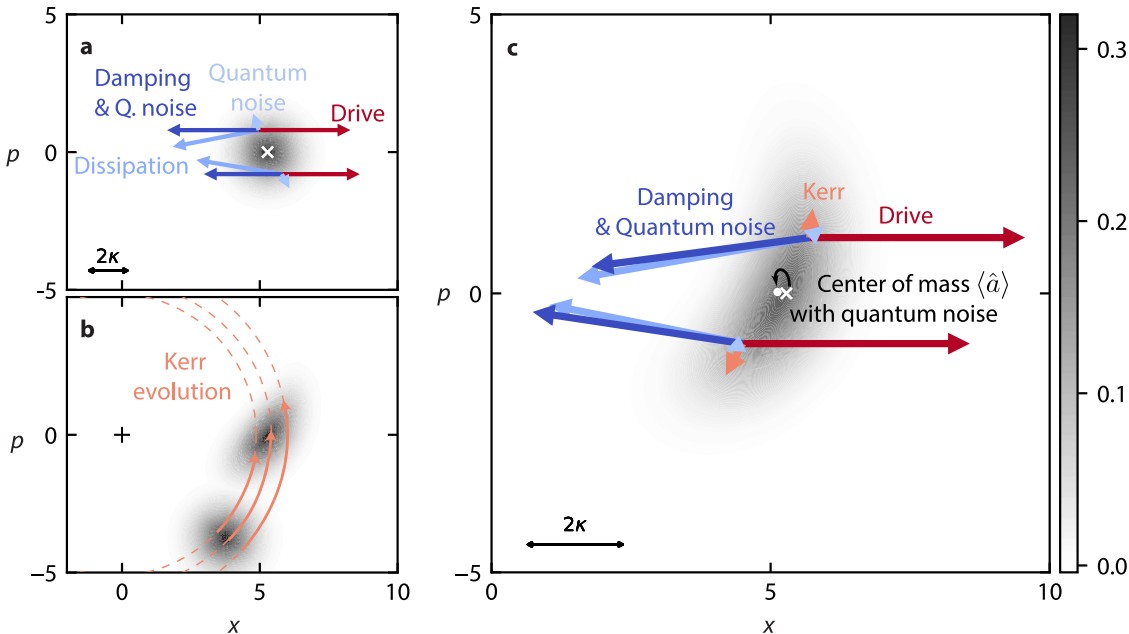

**Fig. 4 | Phase space picture of apparent damping.** Here phase space operators are defined by $\hat{a} = (\hat{x} + i\hat{p})/\sqrt{2}$, see Supplementary Note 5A for further details. **a** Wigner distribution of the steady-state in absence of Kerr nonlinearity (driven at $\omega_r$ with $P_{in} = -124$ dBm). The balance between quantum noise, damping and drive is shown by vectors corresponding to Wigner currents. **b** Growth of phase uncertainty of a coherent state under Kerr nonlinearity. The amplitude-dependent resonance frequency (Kerr effect) translates to a radius-dependent rotation around the origin. The center of mass of the distribution rotates at a frequency $K\alpha^2$. In a frame rotating at that frequency, the effect of the Kerr nonlinearity is to increase the uncertainty in phase (this is the frame adopted in (**c**)). The larger the uncertainty in phase (the extreme case being a ring around the origin), the closer the center of mass of the distribution gets to the origin (i.e., $|\langle\hat{a}\rangle| \to 0$). This is the first contribution (effect A)

to a reduced resonant amplitude's magnitude $|\langle\hat{a}\rangle|$. **c** Wigner distribution of the steady-state with Kerr nonlinearity (at minimum $|S_{21}|$ with $P_{in} = -124$ dBm). The Kerr effect is eventually balanced by the damping, quantum noise and drive. Since the drive now opposes both damping and Kerr effect, it is less effective at opposing the damping and driving the state away from the origin (compared to (**a**)). This brings the distribution closer to the origin, and constitutes the second contribution (effect B) to a reduced resonant amplitude's magnitude $|\langle\hat{a}\rangle|$. The center of mass ($\langle\hat{a}\rangle$) (white dot) is compared to the classical steady-state (white cross). Since the Wigner current of the Kerr effect grows with the amplitude squared $|\langle\hat{a}\rangle|^2 \propto \epsilon^2$ and the drive and dissipation currents grow with $\epsilon$ and $|\langle\hat{a}\rangle|$ respectively, the reduction in $|\langle\hat{a}\rangle|$ does not linearly follow the driving strength $\epsilon$ (see Supplementary Note 5B).

## Quantum mechanical interpretation

We now provide an intuitive explanation as to why there is a decrease in the amplitude of oscillations $\langle\hat{a}\rangle$—leading to an increase in the minimum of $|\langle\hat{S}_{21}\rangle| = |1 - \kappa_{ext}\langle\hat{a}\rangle/(2\epsilon)|$—when quantum noise is considered. Because of the Kerr nonlinearity, the uncertainty in the photon number operator $\hat{a}^\dagger\hat{a}$, translates to uncertainty in the resonance frequency of the oscillator $\omega_r - K\hat{a}^\dagger\hat{a}/2$ (see Hamiltonian of Eq. (1)). This has two consequences. Effect A: the signal leaking out of the oscillator into the transmission line inherits the frequency fluctuations of the oscillator. Since we are measuring a single frequency component with our VNA, we will measure a signal of smaller amplitude. Effect B: the driving is less effective at exciting the oscillator because the resonance frequency of the oscillator is fluctuating and no driving frequency will lead to resonant driving. The average number of photons in the oscillator will decrease.

This interpretation can be more thoroughly explored in phase space, by making use of the Wigner distribution and Wigner current[38–41]. We introduce $\hat{x} = (\hat{a} + \hat{a}^\dagger)/\sqrt{2}$ and $\hat{p} = -i(\hat{a} - \hat{a}^\dagger)/\sqrt{2}$, such that the amplitude $\langle\hat{a}\rangle$ is given by the center of mass of the distribution through $\langle\hat{a}\rangle = \sqrt{2}\iint dx\,dp(x + ip)W$. The Wigner current $\vec{J}$, governs the dynamics of the Wigner function $W$ through the continuity equation $\partial_t W + \vec{\nabla}\vec{J} = 0$. It provides an intuitive visualization of the flow of quasi-probability in phase space.

As a pedagogical starting point, we show in Fig. 4a the distribution and different contributions to the current for a coherent state of amplitude $\alpha$. This state corresponds to the steady-state that would be reached in our resonantly driven system without Kerr nonlinearity. The damping tends to bring each point of the distribution back to the origin. The drive however, is sensitive to phase and acts in a single

direction. These two currents are balanced by the quantum noise, which creates a diffusion of the quasi-probability.

In Fig. 4b, we look at how the Kerr effect deforms the same coherent state, with the damping, driving, and noise temporarily inactive. We see the consequence of the amplitude-dependent resonance frequency of a damped driven quantum Kerr oscillator. In phase space, the resonance frequency sets the rate at which a point rotates around the origin. And the amplitude is given by the distance to the origin. The resulting deformation of the coherent state does not bring any point in phase space closer to the origin (total energy, or photon number, remains constant). The center of mass, however, will move closer to the origin. To be convinced of the latter, one can imagine the extreme case of the Kerr effect deforming the coherent state into a ring circling the origin, so that center of mass would be the origin, and $|\langle\hat{a}\rangle| = 0$. This mechanism for reducing $|\langle\hat{a}\rangle|$ corresponds to Effect A previously discussed.

In the steady-state of our experiment, simulated in Fig. 4c, the evolution of the Kerr effect is eventually balanced by the other currents. Due to the large spread of the state in phase, the diffusion induced by quantum noise is weaker, and the damping current further misaligned with the drive compared to Fig. 4a. Since the drive is not parallel to the combined currents of damping and noise, it is less effective at countering them, so less effective at driving the system. Or in other words, in addition to countering the damping, the drive also has to counter the evolution of the Kerr effect. As a consequence, the average photon-number tends to decrease, which is the second contribution to a lower amplitude (Effect B). In the Supplementary Information (Supplementary Note 5B), we elaborate on why this decrease in amplitude is nonlinear with driving power.

Using a Gaussian state approximation (see Supplementary Note 4), we are able to weigh the influence of Effect A and Effect B in reducing the value of the resonant amplitude $\langle \hat{a} \rangle$. We rely on an analytical comparison of the corresponding amplitude's magnitude $|\langle \hat{a} \rangle|$ and photon number $\langle \hat{a}^\dagger \hat{a} \rangle$, and the fact that Effect A does not affect the photon number, whereas Effect B reduces the amplitude by reducing the photon number from its expected value for a coherent state $\sqrt{\langle \hat{a}^\dagger \hat{a} \rangle} = |\langle \hat{a} \rangle|$. With respect to a coherent state of amplitude $\alpha$, the reduction in $\sqrt{\langle \hat{a}^\dagger \hat{a} \rangle}$ corresponds to half the reduction in $|\langle \hat{a} \rangle|$ in our system (without thermal noise). This means that a reduction in photon number is responsible for only half of the observed effect, indicating that half of the increase in $\text{Min}|S_{21}|$ can be attributed to Effect A, and half to Effect B. The same conclusions can be drawn with thermal noise (assuming $n_{th} \ll \alpha^2$).

While we have focused on the case $n_{th} \ll \alpha^2$, where the damping seems to increase with the amplitude of oscillations, the opposite regime $n_{th} \gg \alpha^2$ has already been explored experimentally[42,43] and bears some common features with this work. When thermal fluctuations dominate, the state of the oscillator is well described as a statistical mixture of oscillatory amplitudes, each shifting the resonance frequency by a different amount given by the Duffing nonlinearity. This results in a broadening of the resonance line-shape when the oscillator is probed, which has been phenomenologically interpreted as an increase in damping, for example in carbon nanotubes[42]. This picture even extends to the case $n_{th} \ll \alpha^2$ where a residual broadening persists due to quantum heating of the oscillator by the driving field[44].

Finally, we note that for all driving strengths featured in our measurements, quadrature squeezing occurs along an axis $u = \cos(\theta)x + \sin(\theta)p$, rotated by an angle $\theta$ with respect the the $x$-axis. At the highest driving power (Figs. 3 and 4), the most highly squeezed quadrature is characterized by $\theta \simeq -0.11\pi$, where the uncertainty $\Delta u$ is 83% of $\Delta x$ for a coherent state.

## Discussion

In conclusion, we have shown how the combination of Kerr nonlinearity and noise, and in particular quantum noise, leads to a dephasing that can manifest in the same way as nonlinear damping. Crucially, our findings are not limited to the case of superconducting resonators. Indeed, preliminary calculations based on our analytical model indicate that this effect has the correct order of magnitude to play a role in the nonlinear damping observed in NEMS systems[14], however, driven by thermal rather than quantum noise. We are therefore confident that this phenomenon can play a valuable role in identifying the nature of nonlinear damping effects in a broader class of systems, such as NEMS or other Josephson circuits, which will be critical to their use in emerging technologies ranging from carbon nanotube sensors to superconducting quantum computing.

## Methods

### Device fabrication

The device shown in Fig. 1 is fabricated in two steps[45]. First, we fabricate the input/output waveguide structures, meandering inductor and capacitor. On a chip of high-resistivity silicon, cleaned in solutions of RCA-1, Piranha, and buffered hydrofluoric acid (BHF), we sputter 60 nm of molybdenum–rhenium (MoRe). A three layer mask (S1813/W(tungsten)/PMMA-950) is then patterned using electron-beam lithography, and is used in etching the MoRe by SF6/He plasma. The mask is finally stripped using PRS 3000.

Secondly, we fabricate the Josephson junctions using the Dolan bridge technique[46]. We first pattern a methyl-methacrylate (MMA)/ polymethyl-methacrylate (PMMA) resist stack with e-beam lithography. After development of the resist, and to ensure a good contact between the aluminum of the junctions and the MoRe, we clean the

sample with an oxygen plasma and BHF. Evaporation of two aluminum layers (30 nm and then 50 nm thick) under two angles (±11 degrees), interposed by an oxidization of the first aluminum layer, forms the junctions. Removal of the resist mask in $N$-methyl-2-pyrrolidone (NMP) at 80 degrees Celsius completes the sample fabrication.

### Data analysis and fitting

Even at lowest driving power, the response of the device does not perfectly fit to a Lorentzian curve, indicating the presence of additional resonances in the measurement chain which could not be calibrated out experimentally. To eliminate these, as well as the change in phase length of the cabling with frequency, we subtract (divide) an affine function of frequency to the measured phase (amplitude).

The transformation between measured response $S_{21,meas}$ and fitted response $S_{21,fit}$ is thus given by

$$(A + B\omega)e^{C + D\omega}S_{21,fit}(\omega) = S_{21,meas} , \tag{5}$$

where $A$, $B$, $C$, and $D$ are determined through a fit of a low-power response, where the nonlinearity does not come into play, and the response of the device alone $S_{21,fit}(\omega)$ is assumed to be

$$S_{21,fit}(\omega) = 1 - \frac{G}{i(\omega - F) + E} . \tag{6}$$

We then reduce the amount of noise as well as the superfluous number of frequency points in the data-set by replacing blocks of 10 successive frequency data-points by their average. The reduction in number of data-points also facilitates the fitting. We were able to numerically compute $S_{21}$ over the 500 frequency points of the data-set in a minimization routine. For each driving power of the data-set, the Python library QuTiP[47,48] was used to solve the Lindblad equation of Eq. (3). For each power, the absolute difference between the 500 (complex) numerical and experimental points constitute a first contribution to the minimized cost function. The difference in the minimum of $|S_{21}|$, and the frequency at which $|S_{21}|$ is minimized, are also added to the cost-function each with a weight of 200 points. The function is minimized using a modified Powell algorithm[49,50], with five free parameters: $\omega_r$, $\kappa_{int}$, $\kappa_{ext}$, $K$, and the attenuation that the signal outputted at the VNA experiences before reaching the device. The attenuation is found to be 118.3 dB, consistent with the physical attenuation installed at room temperature and at the different stages of the dilution refrigerator. The device parameters converge to $\omega_r = 2\pi \times 5.172$ GHz, $\kappa_{int} = 2\pi \times 186$ kHz, $\kappa_{ext} = 2\pi \times 2.12$ MHz, $K = 2\pi \times 80$ kHz.

## Data availability

The data used in this study is available in a Zenodo database with the DOI identifier https://doi.org/10.5281/zenodo.4565179.

## Code availability

The code used to analyze the data and generate all figures is available in a Zenodo database with the DOI identifier https://doi.org/10.5281/zenodo.4565179.

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

## Acknowledgements

We thank M. Kounalakis for discussions. This work was supported by the European Union's Horizon 2020 research and innovation program under grant agreements 681476—QOM3D and 828826—Quromorphic, and by the research program of the Foundation for Fundamental Research on Matter (FOM), which was part of the Dutch Research Council (NWO).

## Author contributions

S.Y. performed the design and fabrication of the device, the measurements, and the initial data-analysis. D.B. and M.F.G. analyzed the data in the context of a classical nonlinear model. M.F.G. and R.v.d.S. analyzed the data using a quantum model, including fitting and the phase space

interpretation, with A.S.M. carrying out the theoretical proof that the quantum model leads to the nonlinear damping equation. M.F.G. wrote the manuscript with contributions from all authors. G.A.S. supervised the project.

## Competing interests

The authors declare no competing interests.
