## [Peer Review File · Nature Communications]

Apparent nonlinear damping triggered by quantum fluctuationsREVIEWER COMMENTS

Reviewer #1 (Remarks to the Author):

Emergence of nonlinear friction from quantum fluctuations

M. F. Gely et al.

In this manuscript, the authors present a theoretical treatment of a Kerr nonlinear resonator in the presence of amplitude fluctuations, as are caused by thermal noise or quantum fluctuations. They show that the combination of the amplitude fluctuations and the energy-conservative nonlinearity leads to an effective nonlinear damping channel. The result of the calculation can explain deviations from a linear drive-amplitude relationship observed in experiments with a Josephson superconducting resonator. The manuscript is written in a pedagogical fashion with step-by-step derivation of the theory results. I can detect no flaw in the methodology and the conclusions appear well supported. This is an interesting piece of work that can be useful to other researchers in several fields. However, I have some questions to the authors before I recommend publication in Nature Communications.

The idea that dephasing due to a combination of a nonlinearity and amplitude noise can lead to nonlinear damping (or more correctly: to deviations from a linear drive-amplitude relationship) is not entirely new. This has for instance already been presented as a potential explanation for the low Q factors of carbon nanotube resonators by Eichler et al. in Nature Commun. 4:2843 (2013). In the present work, the authors show (with a much more complete derivation than in the 2013 paper) that in the limit of zero thermal occupation, the same phenomenon can still take place due to quantum fluctuations. In their equation (4), they even show clearly that the role of quantum fluctuations is not different from that of thermal noise. As they propose, the described mechanism could be responsible for nonlinear damping in many classical devices, which increases the impact of their work considerably. However, the relation to older versions of the same basic idea should be discussed.

On page 7, the authors write “Quantum noise, or the commutation relations of \tilde{a} , \tilde{a}^\dagger , are thus responsible for the change in $\text{Min}(S_{21})$.” This statement seems

to claim that other potential sources of nonlinear damping can be excluded on the basis of the presented material, which I do not think is the case. I ask the authors to consider rephrasing this sentence. I rather agree with their statement on page 10, “In conclusion, we have shown how the combination of Kerr nonlinearity and noise, and in particular quantum noise, can lead to observations of nonlinear damping.”

In general, while the theoretical part of the manuscript is detailed and convincing, the experimental section is short and devoid of any auxiliary measurements. Is there no way to test the hypothesis of nonlinear damping by varying other parameters? Potentially, the authors could inject electrical noise to mimic thermal noise. Also, is there an experimental verification that no significant classical noise enters the device? It would be nice to see a few of these test measurements, which I am sure the authors performed.

Finally, a minor point on terminology. The authors connect quantum fluctuations to nonlinear damping, a word which for me implies energy dissipation, or “force of friction” as in the introduction. However, the explanation for the observed deviations from a linear drive-amplitude relationship is not connected to energy dissipation at all, but to dephasing and, partially, to lack of experimental control. Maybe it would be worth commenting on this difference.

Reviewer #2 (Remarks to the Author):

The manuscript outlines theory and accompanying experimental evidence for a mechanism for nonlinear friction due to quantum fluctuations. The theory outlines how the self-Kerr rotation rate, combined with the phase space distribution due to quantum fluctuations can contribute to shifting the average driven response $\langle \angle a \rangle$, both in phase and amplitude, consistent with a classical picture of nonlinear damping.

I found the general premise interesting, and had not previously considered how the small phase-space distribution of the zero-point fluctuations might manifest itself under Kerr rotation. I thought the explanation in the main text was helpful, and appreciate the much more detailed discussion in the Supplementary. That being said, I can't, at this moment,

recommend this for publication in Nature Communications, specifically because I think that the experimental verification is missing some important checks.

Currently, the entirety of the experimental evidence is embedded in Figures & 3, comprised of a VNA probe power-dependent resonance shift (observed in transmission) and a plot of the shift in the resonance minima values as a function of the probe powers. The Supplementary Information provides a detailed explanation of how the data is fit to account to imperfections in the measurement, presumably including things like Fano corrections to the lineshape that everyone that does these measurements must usually account for. This corrected data is what it used to check alignment with theory. In terms of quantity of data, it is rather limited, and convolves a few different things: the resonant shift due to self-Kerr, the quality of the Fano fitting process, and an unchecked assumption about ground state cavity occupancy.

I think that the experimental evidence would be far stronger if

1. the Kerr constant was verified in a separate measurement, rather than by simple increases in probe power. This can, for instance, be done by driving strongly off-resonance and measuring the resonance shift at low probe power. This (mostly) deconvolves the power-dependent detuning from the instantaneous circulating power in the resonator and serves as an independent check of the Kerr constant. It could also clearly show enhanced damping while retaining the overall mostly symmetric low power resonator response.
2. Equation 4 clearly shows a dependence of the nonlinear damping coefficient γ on both the quantum noise and the thermal occupancy n_{th} . If this is the case, then there should be a strong dependence of the observed nonlinear damping on device temperature at fixed, low pump power. This may be convolved with other effects, like enhanced damping from turning on thermal quasiparticle generation, but it might also be achieved by injecting known white noise.
3. There are also trivial reasons why one might observe a modulation in the dissipation, including a frequency-dependent environmental admittance as seen from the resonator.

This can, for instance, be due to standing waves in the input/output lines that, depending on the setup can induce observable ripple in transmission over 10-200 MHz scales. The effect of this is a frequency dependent real and imaginary component to the environmental admittance. This mechanism for modulation in external damping could also be ruled out by looking at the SQUID modulation curve, S_{21} vs ω & Φ/Φ_0 .

4. In addition to the previous point, the SQUID could also be used to re-check all of the inferred nonlinear damping at a different Josephson energy/Kerr constant. This, at least, could be an important consistency check.

Lastly, I have a small comment on the experimental setup description, lines 31-37 in the Supplemental Information. There is no indication that isolators are used in the experiment, and the description almost sounds like attenuators were placed in the signal path *after* the device. I'm guessing this is a matter of unclear phrasing, but this is easily remedied with a measurement circuit diagram. If, in fact, there are no isolators in the measurement chain, there could also be issues strong HEMT noise factoring into all of these measurements. I suspect that this is not the case, and the description is just unclear.

For what it's worth, I think that the overall idea is interesting, although I'm not exactly convinced that it's critical to understand for other work. The theory alone could be packaged as a good paper in a more specialized journal. I think that as it stands, more experimental checks are needed to support the story for Nature Communications, however.

Reviewer #3 (Remarks to the Author):

The paper describes a careful experimental study of the response of a nonlinear resonator to a moderately strong resonant drive. The nonlinearity comes from the Josephson junction embedded into the circuit. For the studied drive strength, the circuit is well described by an oscillator with the Duffing (Kerr) nonlinearity, which is limited to quartic terms in the effective potential. The experiments are conducted at low temperatures where the thermal occupation number of the oscillator is small. The major observation is the deviation of the response curve from the conventional Duffing response curve. It is attributed to quantum

fluctuations and described in terms of nonlinear friction.

The nonlinear resonant response of a Duffing oscillator has been extensively studied in the literature, both in the classical and in the quantum regime, and important effects of quantum fluctuations on this response have been predicted and observed (see below). However, the effect described in the present paper has not been discussed, to the best of my knowledge. Given the importance of the system as a major well-characterized quantum system far from thermal equilibrium, the results of the paper are of significant broad interest. Therefore, I believe they deserve to be published in Nature Communications.

However, prior to publication, the manuscript has to be modified. The concept of nonlinear friction as a cause of the observation is misleading, from my point of view, and the calculation presented in the Supplemental Material, which is aimed at substantiating this concept, has serious problems.

It is well-established that quantum fluctuations in a periodically driven oscillator lead to a broadening of the Wigner distribution and the distribution over the Floquet states. The effect looks like heating, although this is not the conventional heating related to the rise of the temperature. This specific quantum heating and the associated escape from a stable vibrational state via quantum activation have been clearly seen in the experiment, see Ong et al, PRL 110, 047001 (2013) and Vijay et al., Rev. Sci. Instr. 80, 111101 (2009). The distribution broadening has been found already in Ref. 31 for $T=0$, as indicated in the paper. In the general case of arbitrary temperatures, where the system lacks detailed balance, quantum heating and quantum activation were studied in a number of theory papers starting with the 1980s, see Dykman, PRE 75, 011101 (2007) and Peano and Dykman, NJP 16, 015011 (2014) for a brief review, and also Goto et al., Sci. Rep. 8, 7154 (2018).

On the experimental side, a major distinction of the interplay of fluctuations and nonlinearity from nonlinear friction is seen in the spectra of nonlinear oscillators. Both effects lead to spectral broadening. However, the former makes the spectra asymmetric, whereas the latter does not. Such asymmetry, which is known theoretically since 1970s, has been well established in the experiments on classical nanoscale Duffing oscillators, cf.

Maillet et al., PRB 96, 165434 (2017) and Amarouchene et al., PRL 122, 183901 (2019).

Conceptually, quantum fluctuations in a driven oscillator have nothing in common with any microscopic process that leads to a nonlinear friction. Mathematically, the analysis in the SM is inconsistent: first the fluctuation-induced correction is assumed small and calculated by the perturbation theory, but then it is used to modify the equation for the leading-order term, Eq. (S56). This equation then acquires the form of the equation with nonlinear friction, which is misleading. One may not assert that the observed effect is nonlinear friction based on such an approach.

Incidentally, the nonlinear response of a driven oscillator with nonlinear friction was studied by Buks and Yurke, PRE 74, 046619 (2006), and maybe before that, too. Detailed numerical studies of the Wigner distribution of a resonantly driven quantum oscillator were done by Katz et al., PRL 99, 040404 (2007), NJP 10, 125023 (2008). This work should have been referred when discussing the distribution.

The effect of quantum heating suggests a simple mechanism of the deviation of the nonlinear response from the standard Duffing response. Think of a particle in an asymmetric potential well. For a finite temperature, the mean position of the particle will be shifted from the minimum of the potential. This is a cartoon of what happens with a driven oscillator in the studied regime of a small friction. The effective Hamiltonian of the oscillator in the rotating frame is asymmetric near the position of the stable state. Therefore quantum heating leads to a displacement from this state, on average, i.e., to a different vibration amplitude. The asymmetry depends on the parameters of the field. I emphasize that quantum heating is the effect of quantum fluctuations in a system driven away from thermal equilibrium. The results of the paper do show a new effect of quantum fluctuations, just not a nonlinear friction.

I recommend a major revision of the paper. Again, the observations themselves are interesting and deserve to be published.

Yours sincerely,

Mark Dykman

Reviewer #4 (Remarks to the Author):

The authors study the response to a harmonic drive of a superconducting resonator that has a very small anharmonicity (Kerr term). They find that when the temperature is sufficiently low, such that the thermal occupation of the resonator is totally negligible ($n < 10^{-6}$), the response of the oscillator is compatible with the presence of a small non-linear dissipation. They then study theoretically the response of a non-linear oscillator to a drive in presence of linear dissipation (κ) and a non-linear non-dissipative term (K). They solve numerically the master equation for the stationary solution and obtain analytical results for the response in some specific perturbative regime. This allows the authors to predict the value of the non-linear dissipative coefficient of their experiment (γ), which agrees very well with the simple non-linear picture. They also perform a more sophisticated fit to obtain the microscopic parameters of the system. They perform the data collection and analysis carefully, and present that in details. The result is convincing. I think that it is very unlikely that the source of the non-linear damping they observe is due to some other effect.

The authors then conclude that the non-linear dissipation is well explained by solving the full quantum equation of motion of the non-linear oscillator in presence of dissipation and fluctuations. The authors give two different ways to interpret the non-linear dissipative term. They also give a detailed and very clear analytical analysis in the supplementary materials that derive explicitly this contribution. The intuitive explanation is based on two main effects, A) the frequency noise dephasing and B) the reduction of the effect of the drive. The argument is clear and convincing.

Equation 4 shows that the dissipation is related to the quantum contribution to the energy fluctuation, or zero-point motion fluctuation. The Wigner picture they provide is more involved. I agree with their interpretation, but it is more difficult and technical to follow the arguments. In any case both arguments are convincing and useful.

To my knowledge this effect was not known, and it is sufficiently large to be relevant in the effort to reduce the dissipation in a quantum system close to the ground state. It may have

important consequences in limiting the quality factors of mechanical or electromagnetic resonators in the quantum regime, and it is very useful for the community that this is clearly identified, as is done in this manuscript. The presentation is extremely clear and the manuscript well written (I have few remarks in the following, but these are minor points). I thus think that the manuscript can be very useful for a wide community and could become a reference paper on the subject, since it provides a neat measurement and theoretical discussion of the effect.

Eq. 4 and the interpretation are thus very convincing, they also carry the risk that the effect could be seen and interpreted as a somewhat simple extension of classical fluctuations that cannot vanish at vanishing temperature due to quantum fluctuations. Effectively forgetting the quantum fluctuations and plugging $n_{th}=1/2$ seems to give the same result. I do not know if there are other changes. My impression is that this kind of arguments are easy once the problem has been correctly solved (as it is the case here). Thus my personal view is that the authors provide a very convincing proof of their claims, and that the results are important to a large community, even if it may be possible to interpret in a simple way their findings.

Comments on the manuscript:

-It is maybe a question of personal taste, but I see no reasons to indicate S_{21} in dB in Fig. 2 and Fig. 3. A standard logarithmic scale would be much more appropriate.

-In Fig.2, it would be very useful to indicate the number of photon occupation at maximum drive for each value of the drive. This information is a bit hidden in the manuscript, it is visible in Fig. 4, from the value of $\langle x \rangle$, but it should be given in the text explicitly.

Page 6, line 104, I think that the value for γ obtained from the fit should be given here.

Page 10 line 185, there is a "|" to be eliminated. I also found the phrasing of lines 185-190 a bit obscure before reading the supplementary materials. After reading that it is now clear, but it may be improved for the reader that would not go to the supplementary.

-In the discussion on the fit the authors say that after the fitting procedure they proceed to a slight manual adjustment of the parameters to obtain an improved visual effect (pages 6

line 92 of supplementary materials). This surprises me, since the fit is done to find the best possible value, and it should not be possible to improve it by hand, or one should change some weighting values of the cost function. For instance, weighting the points with the error-bars.

Reviewer response

Note to all the reviewers:

Prompted by a remark of reviewer #3, we note that the steady-state solution of the nonlinearly damped driven classical Kerr oscillator (section S3), and the approximate steady-state solution of the damped driven quantum Kerr oscillator (section S4), can only be reliable far from the threshold of bistability; see Phys. Rev. A 91, 053850 (2015) or Opt. Express 22, 24010 (2014).

We have estimated the critical drive power above which our circuit may showcase bistability to be -122 dBm. Consequently, we have adjusted our theoretical description developed in sections S3 and S4 and restricted the presented data to powers $P_{in} \leq -124$ dBm which lie well below the bistability threshold. This leads to modifications in the main text of the manuscript, and supplementary sections S3 and S4. Additionally, in section S2 we have now compactly rendered the full dataset of the circuit's transmission coefficient in a new figure to illustrate that data measured with drive powers $P_{in} > -124$ dBm lie indeed in the vicinity of the bistability threshold.

Reviewer #1 (Remarks to the Author):

In this manuscript, the authors present a theoretical treatment of a Kerr nonlinear resonator in the presence of amplitude fluctuations, as are caused by thermal noise or quantum fluctuations. They show that the combination of the amplitude fluctuations and the energy-conservative nonlinearity leads to an effective nonlinear damping channel. The result of the calculation can explain deviations from a linear drive-amplitude relationship observed in experiments with a Josephson superconducting resonator. The manuscript is written in a pedagogical fashion with step-by-step derivation of the theory results. I can detect no flaw in the methodology and the conclusions appear well supported. This is an interesting piece of work that can be useful to other researchers in several fields. However, I have some questions to the authors before I recommend publication in Nature Communications.

We thank the reviewer for a careful reading of the manuscript and positive feedback.

The idea that dephasing due to a combination of a nonlinearity and amplitude noise can lead to nonlinear damping (or more correctly: to deviations from a linear drive-amplitude relationship) is not entirely new. This has for instance already been presented as a potential explanation for the low Q factors of carbon nanotube resonators by Eichler et al. in Nature Commun. 4:2843 (2013). In the present work, the authors show (with a much more complete derivation than in the 2013 paper) that in the limit of zero thermal occupation, the same phenomenon can still take place due to quantum fluctuations. In their equation (4), they even show clearly that the role of quantum fluctuations is not different from that of thermal noise. As they propose, the described mechanism could be responsible for nonlinear damping in many classical devices, which increases the impact of their work considerably. However, the relation to older versions of the same basic idea should be discussed.

We agree that the phenomenon presented by Eichler et al. in 2013 is relevant to this work (and we were not aware of it), however the effect is different to the one we discuss here. They consider the situation where thermal noise is dominant with respect to the coherent amplitude of vibration. In their case the oscillator is in a statistical mixture of different oscillation amplitudes, which combined with an amplitude dependent frequency leads to spectral broadening. The spectral broadening does not have a dependence on the coherent amplitude, contrary to the effect discussed in our work, and therefore does not qualify as the nonlinear damping defined in the first line of our main text.

We do however agree that their work complements ours, as it explores the regime where $n_{th} \gg |\langle a \rangle|^2$, whereas we explore the regime $n_{th} \ll |\langle a \rangle|^2$, and we have cited and commented on this work extensively in one of the new concluding paragraphs.

On page 7, the authors write “Quantum noise, or the commutation relations of \tilde{a} , \tilde{a}^\dagger , are thus responsible for the change in $\text{Min}(S_{21})$.” This statement seems to claim that other potential sources of nonlinear damping can be excluded on the basis of the presented material, which I do not think is the case. I ask the authors to consider rephrasing this sentence. I rather agree with their statement on page 10, “In conclusion, we have shown how the combination of Kerr nonlinearity and noise, and in particular quantum noise, can lead to observations of nonlinear damping.”

We agree on this point, and have rephrased the sentence to:

Quantum noise, or the commutation relations of a, a^\dagger , can lead to the entirety of the change in $\text{Min}[S_{21}]$. Thermal noise could lead to a similar effect, but is expected to be negligible in our experiment (see Supplementary Information).

In general, while the theoretical part of the manuscript is detailed and convincing, the experimental section is short and devoid of any auxiliary measurements. Is there no way to test the hypothesis of nonlinear damping by varying other parameters? Potentially, the authors could inject electrical noise to mimic thermal noise. Also, is there an experimental

verification that no significant classical noise enters the device? It would be nice to see a few of these test measurements, which I am sure the authors performed.

The device is unfortunately no longer operational. It was designed and measured for another purpose, and only a few years later this analysis was carried out following some theoretical insight, explaining why these obvious tests were not performed. However, we have now estimated the amount of classical (thermal noise) entering the device, and found it to be negligible. The manuscript has been adapted accordingly (see added section S7B and figure S12 in supplementary information about thermal noise).

Finally, a minor point on terminology. The authors connect quantum fluctuations to nonlinear damping, a word which for me implies energy dissipation, or “force of friction” as in the introduction. However, the explanation for the observed deviations from a linear drive-amplitude relationship is not connected to energy dissipation at all, but to dephasing and, partially, to lack of experimental control. Maybe it would be worth commenting on this difference.

We agree on this point, which was heavily discussed by Reviewer #3 as well. As a result, we have gone through the manuscript and removed mentions of nonlinear friction or nonlinear damping in favour of either of “dephasing” or “apparent nonlinear damping” (we maintain that the effect - phenomenologically at least - still “looks like” nonlinear damping).

Reviewer #2 (Remarks to the Author):

The manuscript outlines theory and accompanying experimental evidence for a mechanism for nonlinear friction due to quantum fluctuations. The theory outlines how the self-Kerr rotation rate, combined with the phase space distribution due to quantum fluctuations can contribute to shifting the average driven response $\langle a \rangle$, both in phase and amplitude, consistent with a classical picture of nonlinear damping.

I found the general premise interesting, and had not previously considered how the small phase-space distribution of the zero-point fluctuations might manifest itself under Kerr rotation. I thought the explanation in the main text was helpful, and appreciate the much more detailed discussion in the Supplementary. That being said, I can't, at this moment, recommend this for publication in Nature Communications, specifically because I think that the experimental verification is missing some important checks.

Currently, the entirety of the experimental evidence is embedded in Figures & 3, comprised of a VNA probe power-dependent resonance shift (observed in transmission) and a plot of the shift in the resonance minima values as a function of the probe powers. The Supplementary Information provides a detailed explanation of how the data is fit to account to imperfections in the measurement, presumably including things like Fano corrections to the lineshape that everyone that does these measurements must usually account for. This corrected data is what it used to check alignment with theory. In terms of quantity of data, it is rather limited, and convolves a few different things: the resonant shift due to self-Kerr, the quality of the Fano fitting process, and an unchecked assumption about ground state cavity occupancy.

I think that the experimental evidence would be far stronger if

1. the Kerr constant was verified in a separate measurement, rather than by simple increases in probe power. This can, for instance, be done by driving strongly off-resonance and measuring the resonance shift at low probe power. This (mostly) deconvolves the power-dependent detuning from the instantaneous circulating power in the resonator and serves as an independent check of the Kerr constant. It could also clearly show enhanced damping while retaining the overall mostly symmetric low power resonator response.

Whilst we agree with the reviewer that this measurement would make the results more convincing, the device is unfortunately no longer operational. As explained to reviewer #1, the device was designed and measured for another purpose, and only a few years later this analysis was carried out following some theoretical insight, with contributors to the project now working in different jobs. This is the reason why some of these obvious tests were not performed.

We do have, however, a good way of testing the validity of the system parameters extracted from the fit, which we expand upon at the end of section S2 in the supplementary information. We measured a reference oscillator which contained only the inductor and capacitor (the SQUID being replaced by a short). This device was built in the same fabrication run as the device measured for this article. With a few extra logical steps (see section S2), the measurement of the reference oscillator gives us an independent measurement of the circuit parameters from which the Kerr constant can be accurately determined.

2. Equation 4 clearly shows a dependence of the nonlinear damping coefficient γ on both the quantum noise and the thermal occupancy n_{th} . If this is the case, then there should be a strong dependence of the observed nonlinear damping on device temperature at fixed, low pump power. This may be convolved with other effects, like enhanced damping from turning on thermal quasiparticle generation, but it might also be achieved by injecting known white noise.

Again, we agree that this measurement would be a good support for the theoretical results presented here. However, as mentioned above, we are no longer able to carry out measurements on this device.

3. There are also trivial reasons why one might observe a modulation in the dissipation, including a frequency-dependent environmental admittance as seen from the resonator. This can, for instance, be due to standing waves in the input/output lines that, depending on the setup can induce observable ripple in transmission over 10-200 MHz scales. The effect of this is a frequency dependent real and imaginary component to the environmental admittance. This mechanism for modulation in external damping could also be ruled out by looking at the SQUID modulation curve, S_{21} vs ω & Φ/Φ_0 .

This is a measurement we actually performed at the time. We have added an analysis of the flux dependency of the resonator response (at low power) in the last paragraph of the supplementary information to address this point.

4. In addition to the previous point, the SQUID could also be used to re-check all of the inferred nonlinear damping at a different Josephson energy/Kerr constant. This, at least, could be an important consistency check.

We agree with the reviewer that this would be a valuable measurement to perform. Unfortunately, we have neither the data nor the means to acquire this data at the moment.

Lastly, I have a small comment on the experimental setup description, lines 31-37 in the Supplemental Information. There is no indication that isolators are used in the experiment, and the description almost sounds like attenuators were placed in the signal path *after* the device. I'm guessing this is a matter of unclear phrasing, but this is easily remedied with a measurement circuit diagram. If, in fact, there are no isolators in the measurement chain, there could also be issues strong HEMT noise factoring into all of these measurements. I suspect that this is not the case, and the description is just unclear.

Attenuators were placed before the device, not after, and isolators were placed between the device and the HEMT. We have added a circuit diagram to the supplementary information (figure S10) to clarify this point.

For what it's worth, I think that the overall idea is interesting, although I'm not exactly convinced that it's critical to understand for other work. The theory alone could be packaged as a good paper in a more specialized journal. I think that as it stands, more experimental checks are needed to support the story for Nature Communications, however.

We appreciate the value that the reviewer places on our work. We strongly believe that the experimental observations, which confirm the validity of the theory with no free parameters provides a "smoking-gun" convincing experimental demonstration suitable for Nature Communications.

Reviewer #3 (Remarks to the Author):

The paper describes a careful experimental study of the response of a nonlinear resonator to a moderately strong resonant drive. The nonlinearity comes from the Josephson junction embedded into the circuit. For the studied drive strength, the circuit is well described by an oscillator with the Duffing (Kerr) nonlinearity, which is limited to quartic terms in the effective potential. The experiments are conducted at low temperatures where the thermal occupation number of the oscillator is small. The major observation is the deviation of the response curve from the conventional Duffing response curve. It is attributed to quantum fluctuations and described in terms of nonlinear friction.

The nonlinear resonant response of a Duffing oscillator has been extensively studied in the literature, both in the classical and in the quantum regime, and important effects of quantum fluctuations on this response have been predicted and observed (see below). However, the effect described in the present paper has not been discussed, to the best of my knowledge. Given the importance of the system as a major well-characterized quantum system far from thermal equilibrium, the results of the paper are of significant broad interest. Therefore, I believe they deserve to be published in Nature Communications.

We thank the reviewer for his careful reading of the manuscript and positive feedback on our work.

However, prior to publication, the manuscript has to be modified. The concept of nonlinear friction as a cause of the observation is misleading, from my point of view, and the calculation presented in the Supplemental Material, which is aimed at substantiating this concept, has serious problems.

It is well-established that quantum fluctuations in a periodically driven oscillator lead to a broadening of the Wigner distribution and the distribution over the Floquet states. The effect looks like heating, although this is not the conventional heating related to the rise of the temperature. This specific quantum heating and the associated escape from a stable vibrational state via quantum activation have been clearly seen in the experiment, see Ong et al, PRL 110, 047001 (2013) and Vijay et al., Rev. Sci. Instr. 80, 111101 (2009). The distribution broadening has been found already in Ref. 31 for $T=0$, as indicated in the paper.

In the general case of arbitrary temperatures, where the system lacks detailed balance, quantum heating and quantum activation were studied in a number of theory papers starting with the 1980s, see Dykman, PRE 75, 011101 (2007) and Peano and Dykman, NJP 16, 015011 (2014) for a brief review, and also Goto et al., Sci. Rep. 8, 7154 (2018).

All of the above experiments, and largely the theory, are focused on the regime of strong driving above or near bifurcation. Our result is in a completely different regime: weak driving before bifurcation. We therefore feel that linking our experiment to this other large body of work would distract from the central point of the article.

On the experimental side, a major distinction of the interplay of [Duffing non-linearity and] fluctuations and nonlinearity from nonlinear friction is seen in the spectra of nonlinear oscillators. Both effects lead to spectral broadening. However, the former makes the spectra asymmetric, whereas the latter does not.

We understand intuitively how non-linear friction alone would lead to symmetric spectral broadening, and we took this point into account in the decision to remove confusing mentions of non-linear friction in our manuscript.

Such asymmetry, which is known theoretically since 1970s, has been well established in the experiments on classical nanoscale Duffing oscillators, cf. Maillet et al., PRB 96, 165434 (2017) and Amarouchene et al., PRL 122, 183901 (2019).

We understand that the reviewer has brought up these articles to point out the difference between non-linear damping and the interplay of nonlinearity and fluctuations.

We find in the first article mentioned (Maillet et al.) particularly interesting to contrast the regime of high thermal fluctuations with the regime of our work $n_{th} \ll \alpha^2$. We have added a paragraph at the end of the manuscript discussing this work and the relation to our results (a similar point was raised by reviewer #1).

Conceptually, quantum fluctuations in a driven oscillator have nothing in common with any microscopic process that leads to a nonlinear friction. Mathematically, the analysis in the SM is inconsistent: first the fluctuation-induced correction is assumed small and calculated by the perturbation theory, but then it is used to modify the equation for the leading-order term, Eq. (S56). This equation then acquires the form of the equation with nonlinear friction, which is misleading. One may not assert that the observed effect is nonlinear friction based on such an approach.

We pick out two different comments made by the reviewer and address them separately.

1. The inconsistency in the mathematical analysis
2. The fact that our described phenomenon has nothing to do with nonlinear friction

We start addressing 1.:

We would like to thank the referee for pointing out the issue with the calculation leading to our former Eq. (S56) and its solution. We think the problem comes from the use of $\Omega \approx -K|\langle a \rangle|^2$, which originally presets the detuning to be $\Delta \approx K|\langle a \rangle|^2 + K/2$.

And this value, which is necessary to derive our previous Eqs. (S54), (S55) and (S56), can be rather different from the value $\Delta = 2Kn_{th} + K(|\langle a \rangle|^2 + 4(K^2/\kappa^2) |\langle a \rangle|^4(n_{th} + \frac{1}{2}))$

that we use soon after to solve our former Eq. (S56) while matching the resonance condition, i.e., while cancelling the term multiplied by the imaginary unit in the left hand side of this previous Eq. (S56). Assigning values that can be appreciably distinct to the same parameter at different stages of the calculation process of a variable's equation solution is certainly mathematically inconsistent. Fortunately, this step in our analysis is unnecessary to consistently derive an equation describing a nonlinearly damped driven Kerr oscillator. We have written down this equation alongside its derivation in the supplementary information (SI) of our manuscript. The equation reads now Eq. (S57) and the steps to derive it are detailed in part F of section S4 of the SI, which we have almost fully rewritten. Therein, we have also amended our former Eq. (S58). This equation provided a formula for the on-resonance value of $\langle a \rangle$ rather than a formula for the on resonance value of $|\langle a \rangle|$. Now we apply Cardano's method to properly find an estimate of the resonant $|\langle a \rangle|$. Likewise, in the discussion of part G of the same section S4 we provide a more straightforward way to arrive at the estimate for the on-resonance photon number based on its corresponding equation of motion.

Additionally, to further support the consistency of the mathematical analysis we carry out for determining an approximate steady state solution of the absolute value of the ensemble average (quantum expectation value) of the oscillator's canonical operator, we now have included the references Phys. Rev. A 91, 053850 (2015), Opt. Exp. 22, 24010 (2014), and Applied Sciences 8, 1427 (2018), our new refs. 29, 52 and 51, respectively. From these references one can notice that our approach amounts to consider a Gaussian state ansatz for the quantum state of the oscillator. This is a fairly good assumption given the regime of moderate driving and weak anharmonicity under which we describe our oscillator.

In its simplest form, such a Gaussian state approximation allows for a linearized description of the system dynamics, in which every operator is approximated by small quantum fluctuations around an ensemble average, the dynamics of which are entirely governed by

the corresponding classical equations of motion of the system. That is, a description of the quantum state of the system based on a particular Gaussian state ansatz: a coherent state. However, given that a Gaussian state can be fully described in terms of the first and second statistical moments of the system's canonical operators, a more general Gaussian state approximation enables a fuller description of the impact that the quantum fluctuations may cause on the ensemble averages of the system's observables. Indeed, using Gaussian statistics (Wick's probability theorem) and the canonical commutation relations, one can write down a closed system of coupled differential equations involving only first order statistical moments (i.e., ensemble averages) and second order statistical moments (covariances) of the system's canonical operators. This approach, differently from the simple coherent state ansatz mentioned above, does not prescribe an operator's ensemble average to be entirely ruled by the classical dynamics of the system. Instead, in this case an operator's ensemble average is treated as an unknown variable to be determined self-consistently together with its covariances, thus adding some information of the quantum dynamics of the system into that of the operator's ensemble averages.

Generally, the closed system of coupled differential equations for ensemble averages and covariances of the system's canonical operators resulting from this Gaussian state approximation is nonlinear, and so normally a solution of it can only be found numerically. Nonetheless, given that any nonlinear term present in these equations shall stem from fourth or higher order statistical moments, far from the threshold of bistability, one may neglect such terms to approximately solve the equations, please see Opt. Exp. 22, 24010 (2014). In our eyes, this approach is consistent and is exactly the approach we have implicitly adopted in our analysis. However, as evidenced by the reviewer's concerns and the more comprehensive explanation shown in the new refs. 29, 51 and 52, we also think our original terminology lacked of clarity and accuracy. Hence, besides the incorporation of these new references we have also refined our explanations accordingly in the introductory part as well as in part A of section S4 of the SI.

Next we address 2.:

Upon reflection and examination of the arguments made by the referee, we agree that the use of "nonlinear friction" is not appropriate to describe our effect. We have gone through the manuscript and removed mentions of nonlinear friction or nonlinear damping in favour of either of "dephasing" or "**apparent** nonlinear damping" (we maintain that the effect - phenomenologically at least - still "looks like" nonlinear damping from the perspective of the expectation value of the field amplitude, as observed experimentally and derived analytically).

Incidentally, the nonlinear response of a driven oscillator with nonlinear friction was studied by Buks and Yurke, PRE 74, 046619 (2006), and maybe before that, too.

A citation to this work has been added in the introduction.

Detailed numerical studies of the Wigner distribution of a resonantly driven quantum oscillator were done by Katz et al., PRL 99, 040404 (2007), NJP 10, 125023 (2008). This work should have been referred when discussing the distribution.

These articles indeed discuss both Wigner distributions and nonlinear driven oscillators. However, we do not feel that this common ground warrants referencing these publications. Indeed, their work looks at the bifurcation regime of oscillators, and therefore the phenomenon of interest is different to the one we are looking at. We feel that the four citations given at the beginning of the discussion on Wigner distributions are more relevant and therefore will provide the reader with a more focused set of literature.

The effect of quantum heating suggests a simple mechanism of the deviation of the nonlinear response from the standard Duffing response. Think of a particle in an asymmetric potential well. For a finite temperature, the mean position of the particle will be shifted from

the minimum of the potential. This is a cartoon of what happens with a driven oscillator in the studied regime of a small friction. The effective Hamiltonian of the oscillator in the rotating frame is asymmetric near the position of the stable state. Therefore quantum heating leads to a displacement from this state, on average, i.e., to a different vibration amplitude. The asymmetry depends on the parameters of the field. I emphasize that quantum heating is the effect of quantum fluctuations in a system driven away from thermal equilibrium. The results of the paper do show a new effect of quantum fluctuations, just not a nonlinear friction.

Again, we agree with the reviewer on this point, nonlinear friction is not a valid way of describing the effect - even though phenomenologically the measurement could be interpreted as such. We have gone through the manuscript and removed mentions of nonlinear friction or nonlinear damping in favour of either “dephasing” or “apparent nonlinear damping” as we maintain that the effect - phenomenologically at least - still “looks like” nonlinear damping.

I recommend a major revision of the paper. Again, the observations themselves are interesting and deserve to be published.

Yours sincerely,

Mark Dykman

The authors thank Mark Dykman for his kind words, as well as his valuable input on our results.

Reviewer #4 (Remarks to the Author):

The authors study the response to a harmonic drive of a superconducting resonator that has a very small anharmonicity (Kerr term). They find that when the temperature is sufficiently low, such that the thermal occupation of the resonator is totally negligible ($n < 10^{-6}$), the response of the oscillator is compatible with the presence of a small non-linear dissipation. They then study theoretically the response of a non-linear oscillator to a drive in presence of linear dissipation (κ) and a non-linear non-dissipative term (K). They solve numerically the master equation for the stationary solution and obtain analytical results for the response in some specific perturbative regime. This allows the authors to predict the value of the non-linear dissipative coefficient of their experiment (γ), which agrees very well with the simple non-linear picture. They also perform a more sophisticated fit to obtain the microscopic parameters of the system. They perform the data collection and analysis carefully, and present that in details. The result is convincing. I think that it is very unlikely that the source of the non-linear damping they observe is due to some other effect.

The authors then conclude that the non-linear dissipation is well explained by solving the full quantum equation of motion of the non-linear oscillator in presence of dissipation and fluctuations. The authors give two different ways to interpret the non-linear dissipative term. They also give a detailed and very clear analytical analysis in the supplementary materials that derive explicitly this contribution. The intuitive explanation is based on two main effects, A) the frequency noise dephasing and B) the reduction of the effect of the drive. The argument is clear and convincing.

Equation 4 shows that the dissipation is related to the quantum contribution to the energy fluctuation, or zero-point motion fluctuation. The Wigner picture they provide is more involved. I agree with their interpretation, but it is more difficult and technical to follow the arguments. In any case both arguments are convincing and useful.

To my knowledge this effect was not known, and it is sufficiently large to be relevant in the effort to reduce the dissipation in a quantum system close to the ground state. It may have important consequences in limiting the quality factors of mechanical or electromagnetic resonators in the quantum regime, and it is very useful for the community that this is clearly identified, as is done in this manuscript. The presentation is extremely clear and the manuscript well written (I have few remarks in the following, but these are minor points). I thus think that the manuscript can be very useful for a wide community and could become a reference paper on the subject, since it provides a neat measurement and theoretical discussion of the effect.

Eq. 4 and the interpretation are thus very convincing, they also carry the risk that the effect could be seen and interpreted as a somewhat simple extension of classical fluctuations that cannot vanish at vanishing temperature due to quantum fluctuations. Effectively forgetting the quantum fluctuations and plugging $n_{th}=1/2$ seems to give the same result. I do not know if there are other changes. My impression is that this kind of arguments are easy once the problem has been correctly solved (as it is the case here). Thus my personal view is that the authors provide a very convincing proof of their claims, and that the results are important to a large community, even if it may be possible to interpret in a simple way their findings.

We thank the reviewer for the positive feedback and for carefully reading the manuscript.

Comments on the manuscript:

-It is maybe a question of personal taste, but I see no reasons to indicate S_{21} in dB in Fig. 2 and Fig. 3. A standard logarithmic scale would be much more appropriate.

We think this is indeed a question of taste. Since it is standard to display S_{21} curves in dB on scientific apparatus, it is our belief that many readers will have a more direct understanding of the plot with this unit.

-In Fig.2, it would be very useful to indicate the number of photon occupation at maximum drive for each value of the drive. This information is a bit hidden in the manuscript, it is visible in Fig. 4, from the value of $\langle x \rangle$, but it should be given in the text explicitly.

We have now added this information in the caption of Fig. 2.

Page 6, line 104, I think that the value for gamma obtained from the fit should be given here.

This information has now been added.

Page 10 line 185, there is a "|" to be eliminated.

This has been addressed.

I also found the phrasing of lines 185-190 a bit obscure before reading the supplementary materials. After reading that it is now clear, but it may be improved for the reader that would not go to the supplementary.

We reworded the paragraph in an attempt to clarify the explanations.

-In the discussion on the fit the authors say that after the fitting procedure they proceed to a slight manual adjustment of the parameters to obtain an improved visual effect (pages 6 line 92 of supplementary materials). This surprises me, since the fit is done to find the best possible value, and it should not be possible to improve it by hand, or one should change some weighting values of the cost function. For instance, weighting the points with the error-bars.

We have re-generated all the figures with the parameters originally determined by the minimization routine, and found that the quality of agreement to data is not really deteriorated.

Although this may be too technical for the current discussion -- we have made the following changes to the fitting routine detailed in the code shared on Zenodo:

Two one line of code in the "cluster_fit.ipynb" file were changed, from

```
1 # converged too:
2 # res_conv = [0.05573881, 2.239268 , 0.91963531, -8.01900103, -0.1142454]
3 # res_conv = [0.05573881, 2.239268 , 0.918, -7.6, 0.07] # manual adjustment
```

To

```
1 # converged too:
2 # res_conv = [0.05573881, 2.239268 , 0.91963531, -8.01900103, -0.1142454]
3 # res_conv = [0.05573881, 2.239268 , 0.918, -7.6, 0.07] # manual adjustment
```

In words rather than code: The manual adjustment line 3 was active in the original submission and now in the revised version line 2 is active and line 3 inactive

This file can be found on our open access zenodo directory for this article (<https://zenodo.org/record/4565179#.YcSrOGjMKMo>). As this version of the manuscript becomes available, we will update our files on zenodo.

The values of the fitted parameters in the text have been adjusted accordingly.

REVIEWER COMMENTS

Reviewer #2 (Remarks to the Author):

I have gone through the authors' response to all referees carefully. They have responded to my questions/concerns carefully. I appreciate the authors' being candid about the fact that the device in this experiment is no longer available for measurement, but that fact is actually important as it allows the reader to take the data with an appropriate grain of salt. I think this might, for instance, be included in the Supplementary. As an experimentalist, I understand how these circumstances arise and that it should also not prevent publication, but I also believe that the appropriate caveats should be noted for the reader. I noticed that the other (presumably) experimentalist referee noted that more convincing support could be provided by measurements similar to the ones that I noted, indicating that these 'missing' measurements will be noted by others.

To summarize, I encourage more transparency about why there is limited measurement data (somewhere-- it doesn't have to be in the main text), but with that small reservation, I otherwise think that their response is suitable for publication.

Reviewer #3 (Remarks to the Author):

The paper has been very significantly modified, starting with the title. This is essentially a new paper. The major error of associating the observation with nonlinear friction has been corrected. I think the observation is interesting and important; it deserves to be published in Nature Communication, which will give it due visibility. However, I believe more changes have to be made prior to publication.

The effect is now described in terms of the currents of the Wigner distribution. I do not find this description physically clear and mathematically unambiguous. A major problem is that it does not explain nor even address the broadening of the spectrum. The separation of the effects into A and B is confusing, given that what is measured is the area under the delta-peak of the spectrum of the periodically driven resonator, it is one quantity, not two. On the formal side, the equation for the Wigner function is well-known (see the references in the

first report and papers cited therein), it is a third-order partial differential equation, and the authors have to solve a hard problem of proving that their separation of the different components of the current is unique. I will address other technical problems in the Supplemental Material further below.

I am confused why the authors avoid the explanation of their observation in the terms familiar from the previous theoretical and experimental work. It was explicitly given in the previous report. Again: the energy levels of a nonlinear mode are nonequidistant. Therefore, when several levels are occupied, the response spectrum is broadened, since the drive can be in resonance with different frequencies. For low temperatures, the occupation of the levels is determined by the effect of quantum heating due to the drive. The observation of quantum heating has nothing to do with the bistability in a strong drive. I would also like to note that the analog of Eq. (4) has been well-known in the weak-drive limit (cf. papers cited in Ref. 40); because of the drive, due to quantum heating, the thermal occupation number of the weak-drive limit has to be replaced by a factor that depends on the drive amplitude and is known from the literature to be proportional to $(1+2n_{\text{th}})$.

On the technical side: the SM relies on the assumption that the Kerr parameter (multiplied by $2n_{\text{th}}+1$) is small compared to the decay rate. Otherwise the decoupling in Eqs. (S28)-(S58) is questionable, to put it gently. But then in Eq. (58) they are allowed to keep only the leading order term in the corresponding small parameter, that is, as it stands, Eq. (S58) is incorrect. I strongly doubt that their approximation is correct beyond the above limit. It is well-known to be wrong for weak drive, in particular. If the authors want to persuade me, they have to calculate higher moments from the full equation and relate them to the lower moments. Otherwise the Gaussian approximation is a postulate, and I don't like postulates at the level of solving an equation.

Also, I don't see why the authors would not use the result of Ref. 33 to describe their observations numerically?

Two remarks on the text:

I would clean up the text a bit. For example, "Quantum noise, or the commutation relations

of a, a^\dagger , can lead..." sounds odd: quantum noise comes from the coupling to a medium, not from the commutation relation. Also, in Eq. (4) the expression $\langle [a, a^\dagger] \rangle$ is odd: the commutator is equal to one by construction, and its average value is one.

In their reply the authors refer to papers 51 and 52. Where can I find them? This is minor.

In conclusion, I want to reiterate that the paper is interesting and important, but it needs further work.

Please find below a point-by-point response to the reviewers' comments.
The reviewers' comments are written in black, whilst the authors' response is written in blue

Reviewer #2

I have gone through the authors' response to all referees carefully. They have responded to my questions/concerns carefully. I appreciate the authors' being candid about the fact that the device in this experiment is no longer available for measurement, but that fact is actually important as it allows the reader to take the data with an appropriate grain of salt. I think this might, for instance, be included in the Supplementary. As an experimentalist, I understand how these circumstances arise and that it should also not prevent publication, but I also believe that the appropriate caveats should be noted for the reader. I noticed that the other (presumably) experimentalist referee noted that more convincing support could be provided by measurements similar to the ones that I noted, indicating that these 'missing' measurements will be noted by others.

To summarize, I encourage more transparency about why there is limited measurement data (somewhere-- it doesn't have to be in the main text), but with that small reservation, I otherwise think that their response is suitable for publication.

We appreciate the reviewers understanding of our circumstances. We have now outlined some of the useful measurements that the reviewer had suggested to confirm our results -- with an explanation of the reason we did not carry these out -- in the supplementary information under the section "Data processing and fitting":

"Finally, we would like to acknowledge that further measurements could have been performed to validate both the system parameters and the physical mechanism of apparent non-linear damping. The Kerr constant could be verified by strongly driving the system off-resonance and measuring the resulting Stark shift with a low-power probe. The effective non-linear damping parameter γ could be verified through the change in S11 expected from a varied thermal occupation n_{th} . Here thermal occupation could be varied by injecting a known white noise or by increasing the base temperature of the dilution refrigerator. Lastly, the SQUID tunability could be used to demonstrate an understanding of the apparent non-linear damping with a different Kerr constant and resonator frequency. Unfortunately, such measurements can no longer be carried out in a practical timescale: the device was fabricated for a different purpose and measured long before the analysis presented in this work was performed and is no longer in working condition."

Reviewer #3

The paper has been very significantly modified, starting with the title. This is essentially a new paper. The major error of associating the observation with nonlinear friction has been corrected. I think the observation is interesting and important; it deserves to be published in Nature Communication, which will give it due visibility. However, I believe more changes have to be made prior to publication.

The effect is now described in terms of the currents of the Wigner distribution. I do not find this description physically clear and mathematically unambiguous.

Whilst this is probably unintentional, the reviewer gives the impression here that the description in terms of Wigner currents is a new addition to the manuscript. It has, however, been present since the first submitted version.

A major problem is that it does not explain nor even address the broadening of the spectrum. The separation of the effects into A and B is confusing, given that what is measured is the area under the delta-peak of the spectrum of the periodically driven resonator, it is one quantity, not two.

The apparent nonlinear damping manifests in multiple ways, one is indeed the broadening of the spectrum, another is the increase in the minimum of $|S_{11}|$. The latter is the feature we have chosen to highlight from our data (Fig2b, Fig3b). Consistently with this choice, we have made the choice to use the Wigner representation to intuitively explain the physical origin of the increase. Unfortunately, it seems this has not provided the insight we expected to the reviewer, although the authors have found it useful to further their intuitive understanding of the effect. Given our approach to analysing the data, we feel it is not strictly necessary for the Wigner representation to provide insight into all the manifestations of this apparent nonlinear damping, namely the spectral broadening.

We note however that it would be good to mention the broadening of the spectrum, using the intuitive picture provided by driving non-equidistant energy levels, as mentioned in the reviewers comment below. Details on incorporating this additional explanation is given below. The Wigner picture is however an inherent part of this article as it stands and we still feel it is a valuable way of intuitively understanding the effect.

On the formal side, the equation for the Wigner function is well-known (see the references in the first report and papers cited therein), it is a third-order partial differential equation, and the authors have to solve a hard problem of proving that their separation of the different components of the current is unique.

The references provided in the first report (authored by Katz et al.) provide an equation for the Wigner function however not in the rotating frame of the drive. Moving to a rotating frame leads to the more intuitive pictures shown in Figs 4, S6, S7, S8, which is why we re-derived the equation for the Wigner function in the supplementary information. This is explained at the beginning of section S5, which, in this revised version, features a reference to the work of Katz et al.

I will address other technical problems in the Supplemental Material further below. I am confused why the authors avoid the explanation of their observation in the terms familiar from the previous theoretical and experimental work. It was explicitly given in the previous report. Again: the energy levels of a nonlinear mode are nonequidistant. Therefore, when several levels are occupied, the response spectrum is broadened, since the drive can be in resonance with different frequencies. For low temperatures, the occupation of the levels is determined by the effect of quantum heating due to the drive. The observation of quantum heating has nothing to do with the bistability in a strong drive. I would also like to note that the analog of Eq. (4) has been well-known in the weak-drive limit (cf. papers cited in Ref. 40); because of the drive, due to quantum heating, the thermal occupation number of the weak-drive limit has to be replaced by a factor that depends on the drive amplitude and is known from the literature to be proportional to $(1+2n_{th})$.

We acknowledge that some crucial references were missing from the manuscript. References to derivations of the spectrum for very similar systems are now included in the discussion of Eq. (4):

“Similar results were derived for the classical [a] and quantum [b] spectrum of undriven oscillators, and also in work studying the spectrum of a probe field in the presence of a strong pump field [c].”

[a] M. I. Dykman and M. A. Krivoglaз, Classical theory of nonlinear oscillators interacting with a medium, *physica status solidi (b)* 48, 497 (1971)

[b] M. Dykman and M. Krivoglaз, Quantum theory of nonlinear oscillators interacting with a medium, *Zhurnal Eksperimental'noj i Teoreticheskoy Fiziki* 64, 993 (1973).

[c] M. Dykman, Periodically modulated quantum nonlinear oscillators, *Fluctuating nonlinear Oscillators* , 165 (2012).

We also acknowledge the value of the intuitive picture provided by the referee here. It was in fact already partially explained at the end of the original manuscript:

“... When thermal fluctuations dominate, the state of the oscillator is well described as a statistical mixture of oscillatory amplitudes, each shifting the resonance frequency by a different amount given by the Duffing nonlinearity. This results in a broadening of the resonance line-shape when the oscillator is probed ...”

What we were missing is the concept of “quantum heating” by the drive, which we have now introduced at the end of this paragraph as follows:

“This picture even extends to [the low temperature case] where a residual broadening persists due to quantum heating of the oscillator by the driving field [d]”

[d] M. I. Dykman, M. Marthaler, and V. Peano, Quantum heating of a parametrically modulated oscillator: Spectral signatures, *Phys. Rev. A* 83, 052115 (2011).

On the technical side: the SM relies on the assumption that the Kerr parameter (multiplied by $2n_{\text{th}}+1$) is small compared to the decay rate. Otherwise the decoupling in Eqs. (S28)-(S58) is questionable, to put it gently. But then in Eq. (58) they are allowed to keep only the leading order term in the corresponding small parameter, that is, as it stands, Eq. (S58) is incorrect. I strongly doubt that their approximation is correct beyond the above limit. It is well-known to be wrong for weak drive, in particular. If the authors want to persuade me, they have to calculate higher moments from the full equation and relate them to the lower moments. Otherwise the Gaussian approximation is a postulate, and I don't like postulates at the level of solving an equation.

Given the concerns outlined above, we think that our writing in section S4 of the supplementary material (SM) has failed to clearly convey to the reviewer some key aspects of our theoretical derivations. We will try to clarify them here.

To derive the steady-state eq. (57) and its corresponding on-resonance solution, eq. (58), we essentially rely on two assumptions.

The first one is to consider that $\langle d^\dagger d^2 \rangle$ is negligible versus $\langle a^\dagger a^2 \rangle - \langle d^\dagger d^2 \rangle$, that $\langle d^\dagger d^3 \rangle$ is negligible versus $\langle a^\dagger a^3 \rangle - \langle d^\dagger d^3 \rangle$ and that $\langle d^3 \rangle$ is negligible versus $\langle a^3 \rangle - \langle d^3 \rangle$. In S4, Eq. (S29), we had formulated this inaccurately. We have now modified eq. (S29) and the text below it.

Based on this assumption we can approximate $\langle a^\dagger a^2 \rangle$ (eq. S34) and $\langle a^\dagger a^3 \rangle$ (eq. S35) in terms of $\langle a \rangle$, $\langle d^2 \rangle$ and $\langle d^\dagger d \rangle$ (and / or their complex conjugated quantities). This constitutes a Gaussian state approximation or self-consistent linearization, and leads us to a first steady-state equation for $\langle a \rangle$, eq. (S53). As discussed in *Opt. Express* 22, 24010 (2014) and *Phys. Rev. A* 91, 053850 (2015) (both references cited in the manuscript), this is a reliable approach whenever the oscillator response is far from

the bistability threshold. We are aware that an analytical steady-state solution stemming from this approach may lack the exact precision achievable by other methods like the ones developed in J. Phys. Math. Gen. 13, 725 (1980) or J. Opt. B: Quantum Semiclass. Opt. 1, 225 (1999). However, it provides a faithful description of our experimental observations (characterised by small thermal fluctuations, eq. (S26), and a weak anharmonicity, eq. (S27)), especially on resonance.

We then think that the Gaussian state approximation is a well grounded assumption rather than a postulate. Nonetheless, from a purely theoretical perspective, we understand the reviewer's concern on this matter. Consequently, in an additional attached document, Adrián Sanz Mora has provided further verification of this first assumption (self-consistent linearization) from a numerical solution of the master equation (S8). He has also written down therein some intermediate steps used to write eq. (S53) that the reviewer might find helpful.

The second assumption we rely on to derive eq. (57) (from eq. (53)) is to consider that $1/(4\Omega^2/\kappa^2 - 4K^2|a\rangle^4/\kappa^2 + 1)$ is equal to 1 near resonance. Figs. S4(a) and S4(b) in part F of section S4 show that, for the entire range of parameters involved in our analysis, the degree of fulfilment of this second assumption is high. Equation (57) is thus only valid near resonance and eq. (58) follows from solving the “on-resonance” version of eq. (57). Importantly, figs. S4(a) and S4(b) are obtained from a numerical solution of the master equation (S8).

Note also that the use of a perturbation series expansion in terms of a small parameter, namely $\varepsilon = 4K^2/\kappa^2$ ($n + 1/2$), eq. (S59), is done after eq. (S58). And the aim of it is to gain a clearer insight of the behaviour of the oscillator's resonant amplitude via an approximate and more easily tractable analytical form of eq. (S58). We state all this in part F of section S4.

Finally, we agree that the approximations behind the theory presented in the supplementary information may not be guaranteed to the level of theoretical rigour that would be required if they were to play a key central role in the conclusions drawn from the work. For that reason, we have softened the language of the main text to clarify that the only claim of the theory from the supplemental information is that if one makes the assumptions in that derivation, one arrives at an analytical result that is in agreement both the data and the numerical simulations:

“... we derive an analytical formula that captures the behaviour of the steady-state response that is in good agreement with the numerical simulations.”

Also, I don't see why the authors would not use the result of Ref. 33 to describe their observations numerically?

We agree that this would have been more efficient as stated in the main text (after equation (3)). However, we only became aware of Ref 33 after having already carried out the analysis in another way. This is also stated in the main text.

Two remarks on the text:

I would clean up the text a bit. For example, “Quantum noise, or the commutation relations of a, a^\dagger , can lead...” sounds odd: quantum noise comes from the coupling to a medium, not from the commutation relation.

We agree with the reviewer that this sentence is confusing. And indeed the coupling to a medium (environment) is crucial for producing non-deterministic “noise”. To avoid confusion, we agree we should avoid mentioning commutation relations here:
“Quantum noise can therefore lead to the entirety of the change in $\text{Min}|S_{21}|$ “

Also, in Eq. (4) the expression $\langle [a, a^\dagger] \rangle$ is odd: the commutator is equal to one by construction, and its average value is one.

We agree and have modified Eq 4 accordingly.

In their reply the authors refer to papers 51 and 52. Where can I find them? This is minor.

These were indeed incorrectly referenced in our previous response. We were referring to Opt. Exp. 22, 24010 (2014), and Applied Sciences 8, 1427 (2018) cited in the supplementary information.

In conclusion, I want to reiterate that the paper is interesting and important, but it needs further work.

We thank the reviewer for their continued interest in our work.

Testing of the Gaussian steady-state approximation

August 27, 2023

1 Statement of the problem

We obtain the equations of motion of the mean $\langle \hat{a} \rangle$ and covariances $\langle \hat{d}^\dagger \hat{d} \rangle$ and $\langle \hat{d}^2 \rangle$ of the canonical operator \hat{a} , with $\hat{d} = \hat{a} - \langle \hat{a} \rangle$, either from the quantum Heisenberg-Langevin equation (S4) or from the Lindblad master equation (S8). They read

$$\frac{d}{dt} \langle \hat{a}(t) \rangle = -[i\Delta + \kappa/2] \langle \hat{a}(t) \rangle + iK \langle \hat{a}^\dagger(t) \hat{a}^2(t) \rangle + \epsilon, \quad (1)$$

$$\frac{d}{dt} \langle \hat{d}^\dagger(t) \hat{d}(t) \rangle = -\kappa \langle \hat{d}^\dagger(t) \hat{d}(t) \rangle + iK \left[\langle \hat{a}^{\dagger 2}(t) \hat{a}(t) \rangle \langle \hat{a}(t) \rangle - \langle \hat{a}^\dagger(t) \rangle \langle \hat{a}^\dagger(t) \hat{a}^2(t) \rangle \right] + \kappa n_{\text{th}}, \quad (2)$$

$$\frac{d}{dt} \langle \hat{d}^2(t) \rangle = -2[i\Delta + \kappa/2] \langle \hat{d}^2(t) \rangle + i2K \left[\langle \hat{a}^\dagger(t) \hat{a}^3(t) \rangle - \langle \hat{a}^\dagger(t) \hat{a}^2(t) \rangle \langle \hat{a}(t) \rangle + \langle \hat{d}^2(t) \rangle / 2 + \langle \hat{a}(t) \rangle^2 / 2 \right], \quad (3)$$

where ϵ is the strength of the microwave drive, n_{th} the thermal occupation number of the resonator, $\Delta = \omega_r - K - \omega_d$ is the detuning, with ω_r and ω_d the angular frequencies of the resonator and the microwave drive respectively, and κ and K represent, respectively, the total energy dissipation rate and Kerr constant of the resonator. To solve the set of equations (1)–(3) above in the steady-state we carry out what in Opt. Express 22, 24010 (2014) and Phys. Rev. A 91, 053850 (2015) (both references cited in the manuscript) is called a Gaussian state approximation or self-consistent linearization. This consists in approximating third and higher order statistical moments of the canonical operators by terms containing only first or second order statistical moments of the canonical operators. The quantum steady-state of the system can then be fully characterized in terms of $\langle \hat{a} \rangle$, $\langle \hat{d}^\dagger \hat{d} \rangle$ and $\langle \hat{d}^2 \rangle$ (and their complex conjugated quantities). Hence the name Gaussian state approximation.

The terms we have to approximate to this end are $\langle \hat{a}^\dagger \hat{a}^2 \rangle$ (and its complex conjugate) and $\langle \hat{a}^\dagger \hat{a}^3 \rangle$. We do so by assuming that their corresponding fluctuations, $\langle \hat{d}^\dagger \hat{d}^2 \rangle$ and $\langle \hat{d}^\dagger \hat{d}^3 \rangle$, are vanishing quantities, i.e., by considering $\langle \hat{d}^\dagger \hat{d}^2 \rangle \simeq 0$ and $\langle \hat{d}^\dagger \hat{d}^3 \rangle \simeq 0$. Using the definition of \hat{d} , we find the fluctuation terms expand into

$$\langle \hat{d}^\dagger \hat{d}^2 \rangle = \langle \hat{a}^\dagger \hat{a}^2 \rangle - 2\langle \hat{a}^\dagger \hat{a} \rangle \langle \hat{a} \rangle + 2\langle \hat{a}^\dagger \rangle \langle \hat{a} \rangle^2 - \langle \hat{a}^\dagger \rangle \langle \hat{a}^2 \rangle, \quad (4)$$

$$\langle \hat{d}^\dagger \hat{d}^3 \rangle = \langle \hat{a}^\dagger \hat{a}^3 \rangle - 3\langle \hat{a}^\dagger \hat{a}^2 \rangle \langle \hat{a} \rangle + 3\langle \hat{a}^\dagger \hat{a} \rangle \langle \hat{a} \rangle^2 - \langle \hat{a}^\dagger \rangle \langle \hat{a}^3 \rangle + 3\langle \hat{a}^\dagger \rangle \langle \hat{a} \rangle \langle \hat{a}^2 \rangle - 3\langle \hat{a}^\dagger \rangle \langle \hat{a} \rangle^3, \quad (5)$$

where, given that we are just concerned with the steady-state, we drop the time argument from now on. Since $\langle \hat{d}^\dagger \hat{d}^3 \rangle$ depends on $\langle \hat{a}^3 \rangle$ we also need to expand $\langle \hat{d} \rangle$. We find

$$\langle \hat{d}^3 \rangle = \langle \hat{a}^3 \rangle - 3\langle \hat{a}^2 \rangle \langle \hat{a} \rangle + 2\langle \hat{a} \rangle^3. \quad (6)$$

By setting $\langle \hat{d}^\dagger \hat{d}^2 \rangle \simeq 0$, $\langle \hat{d}^\dagger \hat{d}^3 \rangle \simeq 0$ and $\langle \hat{d}^3 \rangle \simeq 0$ in equations (4)–(6) we can solve the quantities $\langle \hat{a}^3 \rangle$, $\langle \hat{a}^\dagger \hat{a}^2 \rangle$ and $\langle \hat{a}^\dagger \hat{a}^3 \rangle$. We find the latter two are given by

$$\langle \hat{a}^\dagger \hat{a}^2 \rangle \simeq 2\langle \hat{d}^\dagger \hat{d} \rangle \langle \hat{a} \rangle + \langle \hat{a}^\dagger \rangle \langle \hat{d}^2 \rangle + |\langle \hat{a} \rangle|^2 \langle \hat{a} \rangle, \quad (7)$$

$$\langle \hat{a}^\dagger \hat{a}^3 \rangle \simeq 3\langle \hat{d}^\dagger \hat{d} \rangle \langle \hat{a} \rangle^2 + 3|\langle \hat{a} \rangle|^2 \langle \hat{d}^2 \rangle + |\langle \hat{a} \rangle|^2 \langle \hat{a} \rangle^2, \quad (8)$$

with $|\langle \hat{a} \rangle|^2 = \langle \hat{a}^\dagger \rangle \langle \hat{a} \rangle$, $\langle \hat{d}^\dagger \hat{d} \rangle = \langle \hat{a}^\dagger \hat{a} \rangle - |\langle \hat{a} \rangle|^2$ and $\langle \hat{d}^2 \rangle = \langle \hat{a}^2 \rangle - \langle \hat{a} \rangle^2$. Equations (7) and (8) are, respectively, Eq. (S34) and Eq. (S35) of section S4 of the supplemental information (SI). Substituting the approximations (7) and (8) into the equations of motion (1)–(3) we arrive to the closed set of coupled equations given by Eq. (S36), Eq. (S44) and Eq. (S50) of section S4 of the SI, namely

$$\frac{d}{dt} \langle \hat{a} \rangle \simeq -\left[i(\Delta - 2K\langle \hat{d}^\dagger \hat{d} \rangle - K|\langle \hat{a} \rangle|^2) + \kappa/2 \right] \langle \hat{a} \rangle + iK\langle \hat{a}^\dagger \rangle \langle \hat{d}^2 \rangle + \epsilon. \quad (9)$$

$$\frac{d}{dt} \langle \hat{d}^\dagger \hat{d} \rangle \simeq -\kappa \langle \hat{d}^\dagger \hat{d} \rangle + iK \left[\langle \hat{d}^{\dagger 2} \rangle \langle \hat{a} \rangle^2 - \langle \hat{a}^\dagger \rangle^2 \langle \hat{d}^2 \rangle \right] + \kappa n_{\text{th}}, \quad (10)$$

$$\frac{d}{dt} \langle \hat{d}^2 \rangle \simeq -2 \left[i(\Delta - 2K|\langle \hat{a} \rangle|^2 - K/2) + \kappa/2 \right] \langle \hat{d}^2 \rangle + i2K\langle \hat{a} \rangle^2 \left[\langle \hat{d}^\dagger \hat{d} \rangle + 1/2 \right]. \quad (11)$$

2 Validity tests

The reliability of the Gaussian state approximation or self-consistent linearization, that is, essentially, the assumption $\langle \hat{d}^{\dagger n} \hat{d}^m \rangle \simeq 0$ if $n + m > 2$ with $n, m \in \mathbb{N}$, is subject to the fulfilment that $\langle \hat{d}^{\dagger n} \hat{d}^m \rangle$ is negligible with respect to $\langle \hat{a}^{\dagger n} \hat{a}^m \rangle - \langle \hat{d}^{\dagger n} \hat{d}^m \rangle$. We may compactly express this through the conditions

$$\left| \text{Re} \left[\langle \hat{d}^{\dagger n} \hat{d}^m \rangle \right] \right| \ll \left| \text{Re} \left[\langle \hat{a}^{\dagger n} \hat{a}^m \rangle - \langle \hat{d}^{\dagger n} \hat{d}^m \rangle \right] \right| \quad \text{and} \quad \left| \text{Im} \left[\langle \hat{d}^{\dagger n} \hat{d}^m \rangle \right] \right| \ll \left| \text{Im} \left[\langle \hat{a}^{\dagger n} \hat{a}^m \rangle - \langle \hat{d}^{\dagger n} \hat{d}^m \rangle \right] \right| \quad \text{if } n + m > 2. \quad (12)$$

This is now our Eq. (S29) of section S4 of the SI. Assuming $\langle \hat{d}^{\dagger n} \hat{d}^m \rangle \simeq 0$ may also be viable if the real and/or imaginary parts of each of the quantities are nearly zero. In figures 1–3 we test conditions (12) via a numerical solution of the Lindblad master equation (S8) of section S1 of the SI. We do so for the entire range of detunings and three different drive powers (the lowest one, the highest one and an intermediate one) covered in the analysis of our measurements. In the numerical calculation $n_{\text{th}} = 0$, and the values of $\omega_r/(2\pi) = 5.172$ GHz, $\kappa/(2\pi) = 2.306$ MHz and $K/(2\pi) = 80$ KHz are the ones calibrated out from the measured data. We observe the conditions are satisfied for all the quantities of concerned nearly everywhere. Specially on resonance, the scenario on which we focus our theoretical analysis. We note that the peaks at some ‘exceptional’ detunings in the plots below occur always when both the numerator and the denominator of the quantities plotted are very close to zero. We obtain similar results for $n_{\text{th}} = 1/2$ (with larger peaks at nearly the same ‘exceptional’ detunings).

Figure 1: Plots of (top row) $|\text{Re}[\langle \hat{d}^\dagger \hat{d}^2 \rangle]|$ the absolute value of the real part of $\langle \hat{d}^\dagger \hat{d}^2 \rangle$ normalized to $|\text{Re}[\langle \hat{a}^\dagger \hat{a}^2 - \hat{d}^\dagger \hat{d}^2 \rangle]|$ the absolute value of the real part of $\langle \hat{a}^\dagger \hat{a}^2 - \hat{d}^\dagger \hat{d}^2 \rangle$ and (bottom row) $|\text{Im}[\langle \hat{d}^\dagger \hat{d}^2 \rangle]|$ the absolute value of the imaginary part of $\langle \hat{d}^\dagger \hat{d}^2 \rangle$ normalized to $|\text{Im}[\langle \hat{a}^\dagger \hat{a}^2 - \hat{d}^\dagger \hat{d}^2 \rangle]|$ the absolute value of the imaginary part of $\langle \hat{a}^\dagger \hat{a}^2 - \hat{d}^\dagger \hat{d}^2 \rangle$ as a function of the drive frequency. Black crosses indicate the value of the plotted functions at the resonance detuning (the detuning at which the maximum of $|\langle \hat{a} \rangle|$ is found). Red curves in the panels of the first column are computed with a drive power $P_{\text{in}} = -135.0$ dBm, green curves in the panels of the second column with $P_{\text{in}} = -129.0$ dBm and blue curves in the panels of the third column with $P_{\text{in}} = -124.0$ dBm.

Figure 2: Plots of (top row) $|\text{Re}[\langle \hat{d}^\dagger \hat{d}^3 \rangle]|$ the absolute value of the real part of $\langle \hat{d}^\dagger \hat{d}^3 \rangle$ normalized to $|\text{Re}[\langle \hat{a}^\dagger \hat{a}^3 - \hat{d}^\dagger \hat{d}^3 \rangle]|$ the absolute value of the real part of $\langle \hat{a}^\dagger \hat{a}^3 - \hat{d}^\dagger \hat{d}^3 \rangle$ and (bottom row) $|\text{Im}[\langle \hat{d}^\dagger \hat{d}^3 \rangle]|$ the absolute value of the imaginary part of $\langle \hat{d}^\dagger \hat{d}^3 \rangle$ normalized to $|\text{Im}[\langle \hat{a}^\dagger \hat{a}^3 - \hat{d}^\dagger \hat{d}^3 \rangle]|$ the absolute value of the imaginary part of $\langle \hat{a}^\dagger \hat{a}^3 - \hat{d}^\dagger \hat{d}^3 \rangle$ as a function of the drive frequency. Black crosses indicate the value of the plotted functions at the resonance detuning (the detuning at which the maximum of $|\langle \hat{a} \rangle|$ is found). Red curves in the panels of the first column are computed with a drive power $P_{\text{in}} = -135.0$ dBm, green curves in the panels of the second column with $P_{\text{in}} = -129.0$ dBm and blue curves in the panels of the third column with $P_{\text{in}} = -124.0$ dBm.

Figure 3: Plots of (top row) $|\text{Re}[\langle \hat{d}^2 \rangle]|$ the absolute value of the real part of $\langle \hat{d}^3 \rangle$ normalized to $|\text{Re}[\langle \hat{a}^3 - \hat{d}^3 \rangle]|$ the absolute value of the real part of $\langle \hat{a}^3 - \hat{d}^3 \rangle$ and (bottom row) $|\text{Im}[\langle \hat{d}^3 \rangle]|$ the absolute value of the imaginary part of $\langle \hat{d}^3 \rangle$ normalized to $|\text{Im}[\langle \hat{a}^3 - \hat{d}^3 \rangle]|$ the absolute value of the imaginary part of $\langle \hat{a}^3 - \hat{d}^3 \rangle$ as a function of the drive frequency. Black crosses indicate the value of the plotted functions at the resonance detuning (the detuning at which the maximum of $|\langle \hat{a} \rangle|$ is found). Red curves in the panels of the first column are computed with a drive power $P_{\text{in}} = -135.0$ dBm, green curves in the panels of the second column with $P_{\text{in}} = -129.0$ dBm and blue curves in the panels of the third column with $P_{\text{in}} = -124.0$ dBm.

3 Step by step derivation of Eq. (S53)

Note: all the calculations shown below use no approximation. The symbol ‘ \simeq ’ is used to indicate that equations derive from the Gaussian state approximation explained above in section 1.

Solving eqs. (10) and (11) for the covariances in the steady-state, we have

$$\langle \hat{d}^\dagger \hat{d} \rangle \simeq \frac{n_{\text{th}} + \frac{1}{2} \frac{4K^2 |\langle \hat{a} \rangle|^4}{4\Omega^2 + \kappa^2}}{1 - \frac{4K^2 |\langle \hat{a} \rangle|^4}{4\Omega^2 + \kappa^2}} = \frac{n_{\text{th}} + \frac{1}{2}}{1 - \frac{4K^2 |\langle \hat{a} \rangle|^4 / \kappa^2}{4\Omega^2 / \kappa^2 + 1}} - \frac{1}{2}, \quad (13)$$

$$\langle \hat{d}^2 \rangle \simeq i \frac{K \langle \hat{a} \rangle^2}{i\Omega + \kappa/2} \left(\langle \hat{d}^\dagger \hat{d} \rangle + \frac{1}{2} \right) = \frac{4K}{\kappa} \langle \hat{a} \rangle^2 \frac{n_{\text{th}} + \frac{1}{2}}{4\Omega^2 / \kappa^2 - 4K^2 |\langle \hat{a} \rangle|^4 / \kappa^2 + 1} (\Omega / \kappa + i/2), \quad (14)$$

where $\Omega = \Delta - K(2|\langle \hat{a} \rangle|^2 + 1/2)$. The second equality of eq. (13) follows from adding and subtracting $1/2$ in the numerator. In the second equality of eq. (14) we have explicitly written $\langle \hat{d}^2 \rangle$ in terms of its real and imaginary parts. Substituting eqs. (13) and (14) into eq. (9) leads to

$$\begin{aligned} 0 \simeq & -i[\Delta + \kappa/2] \langle \hat{a} \rangle + i2K \frac{n_{\text{th}} + \frac{1}{2}}{1 - \frac{4K^2 |\langle \hat{a} \rangle|^4 / \kappa^2}{4\Omega^2 / \kappa^2 + 1}} \langle \hat{a} \rangle - iK \langle \hat{a} \rangle + iK |\langle \hat{a} \rangle|^2 \langle \hat{a} \rangle \\ & + iK |\langle \hat{a} \rangle|^2 \frac{\Omega}{\kappa} \frac{4K}{\kappa} \frac{n_{\text{th}} + \frac{1}{2}}{4\Omega^2 / \kappa^2 - 4K^2 |\langle \hat{a} \rangle|^4 / \kappa^2 + 1} \langle \hat{a} \rangle - \frac{1}{2} K |\langle \hat{a} \rangle|^2 \frac{4K}{\kappa} \frac{n_{\text{th}} + \frac{1}{2}}{4\Omega^2 / \kappa^2 - 4K^2 |\langle \hat{a} \rangle|^4 / \kappa^2 + 1} \langle \hat{a} \rangle + \epsilon. \end{aligned} \quad (15)$$

Next we subtract to the eq. (15) above all the terms proportional to $\langle \hat{a} \rangle$ in its right hand side (rhs) and add them in its left hand side (lhs). Then we group together all the real terms as well as all the imaginary terms that multiply $\langle \hat{a} \rangle$ in the lhs of eq. (15). The outcome reads

$$\begin{aligned} & \left[i \left(\Delta + K - K |\langle \hat{a} \rangle|^2 - 2K \frac{n_{\text{th}} + \frac{1}{2}}{1 - \frac{4K^2 |\langle \hat{a} \rangle|^4 / \kappa^2}{4\Omega^2 / \kappa^2 + 1}} - K |\langle \hat{a} \rangle|^2 \frac{\Omega}{\kappa} \frac{4K}{\kappa} \frac{n_{\text{th}} + \frac{1}{2}}{4\Omega^2 / \kappa^2 - 4K^2 |\langle \hat{a} \rangle|^4 / \kappa^2 + 1} \right) \right. \\ & \left. + \frac{\kappa}{2} \left(1 + \frac{K |\langle \hat{a} \rangle|^2}{\kappa} \frac{4K}{\kappa} \frac{n_{\text{th}} + \frac{1}{2}}{4\Omega^2 / \kappa^2 - 4K^2 |\langle \hat{a} \rangle|^4 / \kappa^2 + 1} \right) \right] \langle \hat{a} \rangle \simeq \epsilon \end{aligned} \quad (16)$$

We can write the eq. (16) above more compactly by noticing that

$$\begin{aligned}
& 2K \frac{n_{\text{th}} + \frac{1}{2}}{1 - \frac{4K^2 |\langle \hat{a} \rangle|^4 / \kappa^2}{4\Omega^2 / \kappa^2 + 1}} + K |\langle \hat{a} \rangle|^2 \frac{\Omega}{\kappa} \frac{4K}{\kappa} \frac{n_{\text{th}} + \frac{1}{2}}{4\Omega^2 / \kappa^2 - 4K^2 |\langle \hat{a} \rangle|^4 / \kappa^2 + 1} = \\
& 2K \frac{n_{\text{th}} + \frac{1}{2}}{4\Omega^2 / \kappa^2 - 4K^2 |\langle \hat{a} \rangle|^4 / \kappa^2 + 1} \left[4\Omega^2 / \kappa^2 + 1 + \frac{\Omega}{\kappa} \frac{2K |\langle \hat{a} \rangle|^2}{\kappa} \right] = \\
& 2K \left(n_{\text{th}} + \frac{1}{2} \right) \left[\frac{4\Omega^2 / \kappa^2 - 4K^2 |\langle \hat{a} \rangle|^4 / \kappa^2 + 1 + \frac{\Omega}{\kappa} \frac{2K |\langle \hat{a} \rangle|^2}{\kappa} + 4K^2 |\langle \hat{a} \rangle|^4 / \kappa^2}{4\Omega^2 / \kappa^2 - 4K^2 |\langle \hat{a} \rangle|^4 / \kappa^2 + 1} \right] = \\
& 2K \left(n_{\text{th}} + \frac{1}{2} \right) \left[1 + \frac{2K |\langle \hat{a} \rangle|^2}{\kappa} \frac{\Omega / \kappa + 2K |\langle \hat{a} \rangle|^2 / \kappa}{4\Omega^2 / \kappa^2 - 4K^2 |\langle \hat{a} \rangle|^4 / \kappa^2 + 1} \right] = \\
& 2K \left(n_{\text{th}} + \frac{1}{2} \right) \left[1 + \frac{2K |\langle \hat{a} \rangle|^2}{\kappa^2} \frac{\Delta - K/2}{4\Omega^2 / \kappa^2 - 4K^2 |\langle \hat{a} \rangle|^4 / \kappa^2 + 1} \right] = \\
& 2K \left(n_{\text{th}} + \frac{1}{2} \right) + \frac{K |\langle \hat{a} \rangle|^2}{\kappa} \frac{4K}{\kappa} \frac{n_{\text{th}} + \frac{1}{2}}{4\Omega^2 / \kappa^2 - 4K^2 |\langle \hat{a} \rangle|^4 / \kappa^2 + 1} (\Delta - K/2) \quad (17)
\end{aligned}$$

Adding and subtracting $iK \langle \hat{a} \rangle / 2$ while using the last equality of eq. (17) in the lhs of eq. (16), results in

$$\begin{aligned}
& i \left[(\Delta - K/2) \left(1 - \frac{K |\langle \hat{a} \rangle|^2}{\kappa} \frac{4K}{\kappa} \frac{n_{\text{th}} + \frac{1}{2}}{4\Omega^2 / \kappa^2 - 4K^2 |\langle \hat{a} \rangle|^4 / \kappa^2 + 1} \right) + 3K/2 - K |\langle \hat{a} \rangle|^2 - 2K \left(n_{\text{th}} + \frac{1}{2} \right) \right] \langle \hat{a} \rangle \\
& + \frac{\kappa}{2} \left(1 + \frac{K |\langle \hat{a} \rangle|^2}{\kappa} \frac{4K}{\kappa} \frac{n_{\text{th}} + \frac{1}{2}}{4\Omega^2 / \kappa^2 - 4K^2 |\langle \hat{a} \rangle|^4 / \kappa^2 + 1} \right) \langle \hat{a} \rangle \simeq \epsilon \quad (18)
\end{aligned}$$

If we introduce the coefficient

$$\gamma = \frac{4K^2}{\kappa} \left(n_{\text{th}} + \frac{1}{2} \right), \quad (19)$$

then eq. (18) reads

$$\begin{aligned}
& i \left[\left(\Delta - \frac{K}{2} \right) \left(1 - \frac{\gamma |\langle \hat{a} \rangle|^2 / \kappa}{4\Omega^2 / \kappa^2 - 4K^2 |\langle \hat{a} \rangle|^4 / \kappa^2 + 1} \right) + \frac{3K}{2} - K |\langle \hat{a} \rangle|^2 - 2K \left(n_{\text{th}} + \frac{1}{2} \right) \right] \langle \hat{a} \rangle \\
& + \frac{\kappa}{2} \left(1 + \frac{\gamma |\langle \hat{a} \rangle|^2 / \kappa}{4\Omega^2 / \kappa^2 - 4K^2 |\langle \hat{a} \rangle|^4 / \kappa^2 + 1} \right) \langle \hat{a} \rangle \simeq \epsilon. \quad (20)
\end{aligned}$$

This is Eq. (S59) in section S4 of the SI. Figure 4 compares a solution of $|\langle \hat{a} \rangle|$ as obtained from Eq. (S53) (equation (20) above) with a solution of $|\langle \hat{a} \rangle|$ as obtained from the Lindblad master equation (S8) (using again $n_{\text{th}} = 0$ and the same parameter values that we specify in section 1). We observe excellent agreement between the two solutions, corroborating further the validity of our Gaussian state approximation. Good agreement is also observed for $n_{\text{th}} = 1/2$.

Figure 4: Comparison between $|\langle \hat{a} \rangle|$ computed from the Lindblad master equation (S8) (full curve) and $|\langle \hat{a} \rangle|$ computed from a solution of eq. (20) (dashed curve). Plots in the left most panel are obtained with $P_{\text{in}} = -135.0$ dBm, plots in the middle panel with $P_{\text{in}} = -129.0$ dBm, and plots in the right most panel with $P_{\text{in}} = -124.0$ dBm. Black crosses indicate the maximum (resonant) value of $|\langle \hat{a} \rangle|$ (obtained from the solution of the Lindblad master equation (S8)).

REVIEWERS' COMMENTS

Reviewer #3 (Remarks to the Author):

The authors have adequately addressed all comments in the reports. I recommend publishing the paper in the present form.